# Clinical features and prognostic factors of IV combined small cell lung cancer: A propensity score matching analysis

Shanshan Cai [ORCID], Weichang Yang, Hongdan Luo, Zhouhua Li, Xiaotian Huang, Jinbo Li, Xiaoqun Ye [ORCID] *

Department of Respiratory and Critical Care Medicine, The Second Affiliated Hospital, Jiangxi Medical College, Nanchang University, Nanchang City, Jiangxi Province, People's Republic of China

* 511201663@qq.com

## Abstract

### Background

Nowadays, the characteristics and treatment of combined small-cell lung carcinoma (CSCLC) remain controversial. This study aimed to analyze the features of clinical demographics, survival outcomes and treatment modalities among IV CSCLC, IV SCLC and IV NSCLC, to provide more evidence for the study of IV CSCLC.

### Methods

All CSCLC, SCLC and NSCLC patient data were obtained from the SEER database (2010–2020). Pearson's $\chi2$ test was used to compare the differences in clinical characteristics. Propensity score matching (PSM) was utilized to balance the bias of the variables between patients. Univariate and multivariate Cox proportional hazards regression analyses were performed to identify prognostic factors. KM analysis was used to calculate survival. Adjusted analyses for the primary outcome of different treatment modalities of IV CSCLC, IV SCLC and IV NSCLC were performed using Cox regression models.

### Results

A total of 493 patients with IV CSCLC, 35503 patients with SCLC, 122807 patients with IV NSCLC were included in this study. The demographic characteristics and tumor characteristics of the three groups were different. Before PSM, there were significant differences in OS and CSS among IV CSCLC, IV SCLC and IV NSCLC, After PSM, there was a significant difference in OS and CSS between the IV CSCLC and IV NSCLC. Risk/protective factors for OS and CSS were different in three groups. Chemotherapy, radiotherapy, and surgery can improve IV CSCLC's survival time. The combination of surgery and chemoradiotherapy treatment group for patients with IV CSCLC demonstrated best OS compared to control treatment groups, and the surgery combined chemotherapy treatment group exhibited the best CSS. Additionally, for select patients with stage IV CSCLC who have missed the

**Data Availability Statement:** Data are retrieved from the Surveillance, Epidemiology, and End Results (SEER) databases and therefore authors cannot share it publicly. When researchers access

SEER data, they enter into a data use agreement (DUA) with National Cancer Institute (https://seer.cancer.gov/index.html) that mandates compliance with these privacy protections. According to this agreement, we are prohibited from redistributing the data to third parties. The terms of the DUA are designed to protect patient confidentiality and to ensure that data is used solely for the purposes outlined in the approved research proposal. Request forms may be accessed at https://seer.cancer.gov/data/access.html.

**Funding:** This study was supported by the National Natural Science Foundation of China (Grant No. 81660493), the Natural Science Foundation of Jiangxi Province (Grant No.20202ACBL206019) and the Jiangxi Province key research and development project (20243BBI91024).

**Competing interests:** The authors have declared that no competing interests exist.

window for surgical intervention at the time of initial diagnosis, chemoradiotherapy presents a viable and effective treatment option.

## Conclusions

The clinical characteristics IV CSCLC, IV SCLC and IV NSCLC were significantly different. The prognosis for IV CSCLC is notably poorer than IV NSCLC, similar to IV SCLC. Surgery combined therapy emerged as the preferred treatment modalities and chemoradiotherapy was a good choice for patients who have lost the indication of surgery for patients diagnosed with IV CSCLC.

## Introduction

Lung cancer is one of the most common cancers and the leading cause of cancer-related deaths worldwide, with an estimated 2 million new cases and 1.76 million deaths per year [1]. Combined small-cell lung carcinoma (CSCLC) is a rare tissue type of lung cancer, the initial definition of CSCLC dates back to 1999 [2], the histological classification of the World Health Organization (WHO) classifies SCLC into SCLC and Combined SCLC (CSCLC), which accounts for about 5–20% of total SCLC cases [3, 4]. CSCLC is a relatively rare subtype of SCLC, which refers to a subtype that has both SCLC histology and any subtype of non-small cell lung cancer (NSCLC) [5].

CSCLC is closely related to the genotypic and immunophenotypic of NSCLC and SCLC. Epidermal growth factor receptor (EGFR) mutations are present in NSCLC and related to tumor response to EGFR tyrosine kinase inhibitors (TKIs), indicating that EGFR constitutes a potential biomarker. Previous studies reported that EGFR mutations occur in less than 5% of SCLC cases, while a rate reaching 15%–20% can be found in CSCLC [6–8]. Wagner et al. first assessed 7 CSCLC cases for genotypic and immunophenotypic associations determining whether NSCLC constituents displayed features specific to SCLC [9]. In this study, several biomarkers were utilized, further indicating that a common clonal precursor with closer relationship with SCLC than NSCLC exists, CSCLCs are closer to SCLC than NSCLC.

It is very meaningful to compare prognosis and clinical features due to the c association of genotypes and immunophenotypes. However, because of the rarity of the CSCLC, little research has been done on the prognosis and treatment of patients with subtypes of CSCLC and comparison with SCLC and NSCLC, especially IV stage. Only several studies compared the survival and prognosis of CSCLC and SCLC [10–13], but conflicting results have been reported by different studies. As for the study about comparison of CSCLC and NSCLC is even rarer, and no relevant studies on the IV stage have been conducted so far.

Most cases of CSCLC belonged to advanced stages when they were firstly diagnosed. Zhang et al. reported almost 90% of CSCLC were diagnosed as stage III and IV in their cohort [14]. Thus, the study of IV CSCLC is of great clinical interest The Surveillance, Epidemiology, and End Results (SEER) database is unique in the number of cases, especially for IV CSCLC patients. In this study, we obtained stage IV CSCLC, IV SCLC and IV NSCLC data from SEER database and performed propensity score matching (PSM) analysis to compare the survival outcome, prognostic factors and treatment modalities of three group.

## Materials and methods

### Data collection

SEER is a United States cancer patient-based database that collects data on approximately 30% of all cancer patients with the goal of reducing the burden of cancer (https://seer.cancer.gov/). CSCLC, SCLC and NSCLC data were downloaded from SEER Research Data, 17 Registries, Nov 2022 Sub (2000–2020). Inclusion criteria: (1) all subjects were diagnosed in 2010–2020; (3)Site and Morphology. CS Schema-AJCC 6th Edition: Lung; (4) histology code (ICD-O-3 Hist/behav): 8045/3, 8002/3, 8041/3, 8042/3, 8043/3, 8044/3, 8012/3, 8070/3, 8071/3, 8072/3, 8073/3, 8074/3, 8075/3, 8076/3 and 8140/3. (5) The tumor stage was IV. Exclusion criteria: (1) Follow-up data unknown and missing; (2) Incomplete clinical data and other relevant information.

The variables collected included demographic characteristics of patients: age, gender, race and marital status. Tumor characteristics: laterality, T stage, N stage, brain metastasis, bone metastasis, liver metastasis, lung metastasis, primary site. Treatment: Surgery, Radiation and chemotherapy. Survival data: Survival months, overall survival (OS) and cancer-specific survival (CSS). To facilitate statistical analysis, we reclassified some variables: age ($\geq 65, < 65$), marital status (married, divorced, others), T stage (T0, T1, T2, T3, T4, TX), N stage (N0, N1, N2, N3, NX), laterality (left, right, others), primary site (main bronchus, upper lobe, middle lobe, lower lobe, others). OS and CSS were the primary endpoints in this study. Patients diagnosed in 2016–2017 were reclassified to T stage and N stage according to the "2016 SEER Manual Section V: Stage of Disease at Diagnosis" document. Definition of Treatment options: (1) Radiotherapy: Yes: patients were treated with radiotherapy as the first course of treatment. No: patients were not treated with radiotherapy as the first course of treatment. (2) Chemotherapy: Yes: patients were treated with chemotherapy. No: patients were not treated with chemotherapy. (3) Surgery: Yes: patients were treated with surgery. No: patients were not treated with surgery. The flow chart of patient screening is shown in Fig 1.

### Propensity score matching

To reduce the effect of selection bias, propensity score matching (PSM) was applied to IV CSCLC, IV SCLC and IV NSCLC groups in this study. The matching ratio for IV CSCLC and IV SCLC groups was 1:4 and the caliper value was set to 0.02 through the "nearest" method (Fig 2A), the same as the matching ratio (1:4) and caliper value (0.02) for IV CSCLC and IV NSCLC groups (Fig 2B). The variables used for matching were as follows: age, gender, race, marital status, T stage, N stage, laterality, primary site, brain metastasis, bone metastasis, liver metastasis, lung metastasis, surgery, radiotherapy, chemotherapy.

Similarly, PSM was applied to each treatment modality in IV CSCLC patients in this study. For surgery vs. no surgery groups, a 1:4 matching ratio with a 0.02 caliper value was adopted using the "nearest" method (S1A Fig and S3 Table). Similarly, 1:1 ratio and 0.02 caliper values were applied to radiotherapy vs. no radiotherapy groups (S1B Fig and S4 Table) and chemotherapy vs. no chemotherapy groups (S1C Fig and S5 Table), also utilizing the "nearest" method. Matching variables encompassed age, gender, race, marital status, T stage, N stage, laterality, primary site, brain metastasis, bone metastasis, liver metastasis, lung metastasis, along with other therapies received (surgery /radiotherapy/ chemotherapy).

### Statistical methods

All statistical analyses were performed with SPSS 23.0(SPSS Inc., Chicago, IL, USA) and R version 4.2.1. P-value < 0.05 was considered statistically significant. The "MatchIt" "tableone"

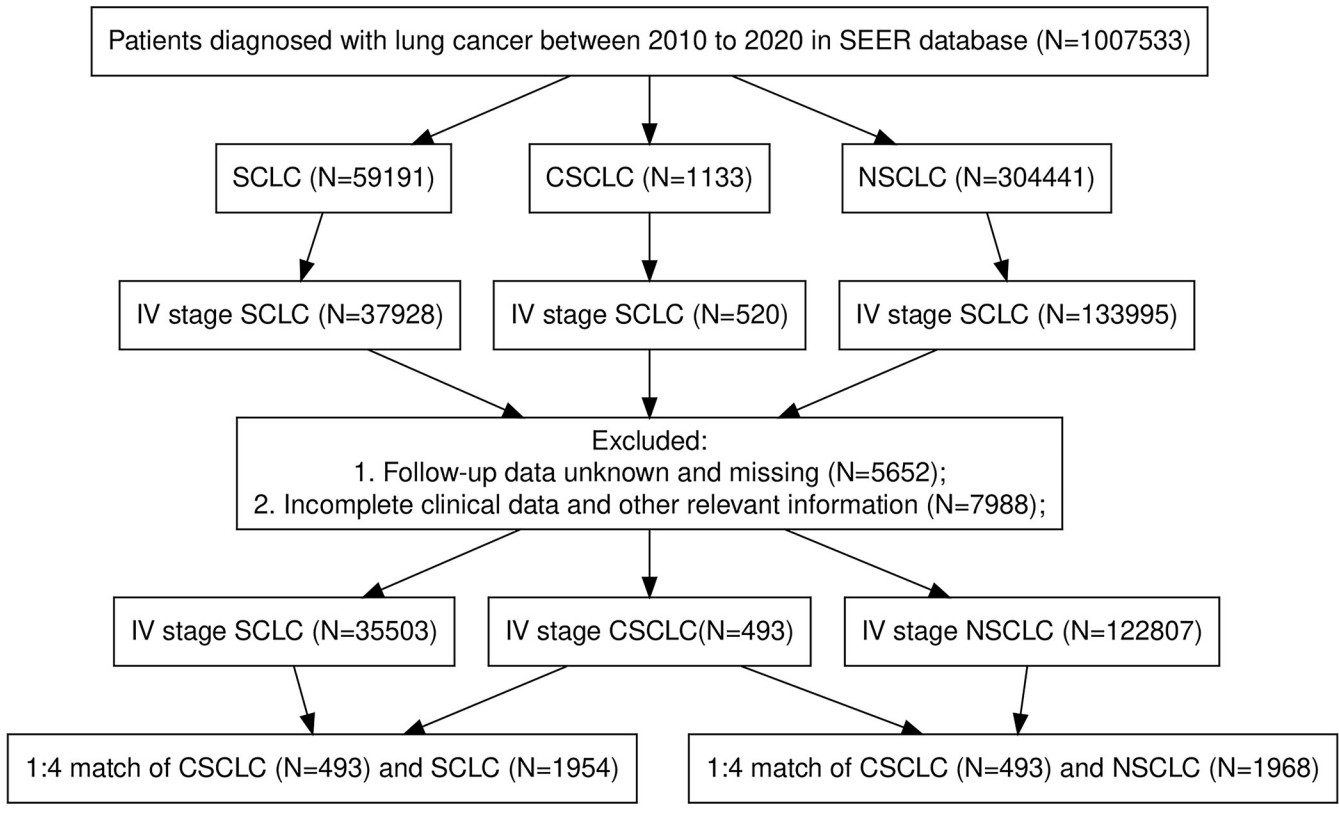

**Fig 1. Flow chart for screening patients.**

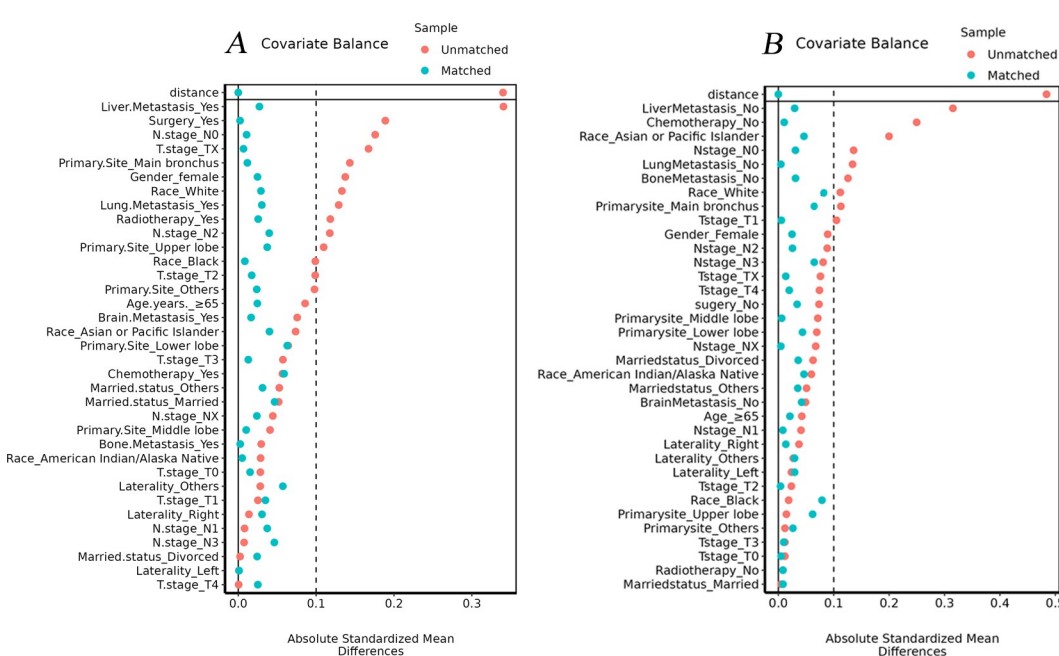

**Fig 2. Standardized mean differences before and after PSM.** A: 1:4 match of CSCLC and SCLC; B: 1:4 match of CSCLC and NSCLC.

package of R was used to perform PSM analyses and Standardized mean difference (SMD). Kaplan–Meier method and log-rank test were used to compare the prognosis of different groups and treatment modalities. For comparison between groups of categorical data, we used the Fisher exact test for expected frequencies of <5, otherwise, we used the Chi-squared test. Univariable and multivariate Cox proportional hazard models were used to identify risk factors for OS and CSS in the three groups. Adjusted analyses for the primary survival outcome of different treatment modalities of IV CSCLC, IV SCLC and IV NSCLC were performed using Cox regression models to estimated hazard ratios (HRs) with corresponding two sides 95CIs, considering potential unbalanced confounders that may have influenced the outcomes.

## Result

### Epidemiology for IV CSCLC, IV SCLC and IV NSCLC

The semi-logarithmic line chart was used to describe the number of cases per year from 2000 to 2020 per year in the three groups. It can be seen in the figure that the number of patients in NSCLC was the highest, followed by SCLC, and the number of patients in CSCLC was the lowest. The number of patients in NSCLC kept increasing from 2004 to 2017; but was generally stable. However, the incidence of CSCLC and SCLC did not fluctuate significantly (Fig 3A).

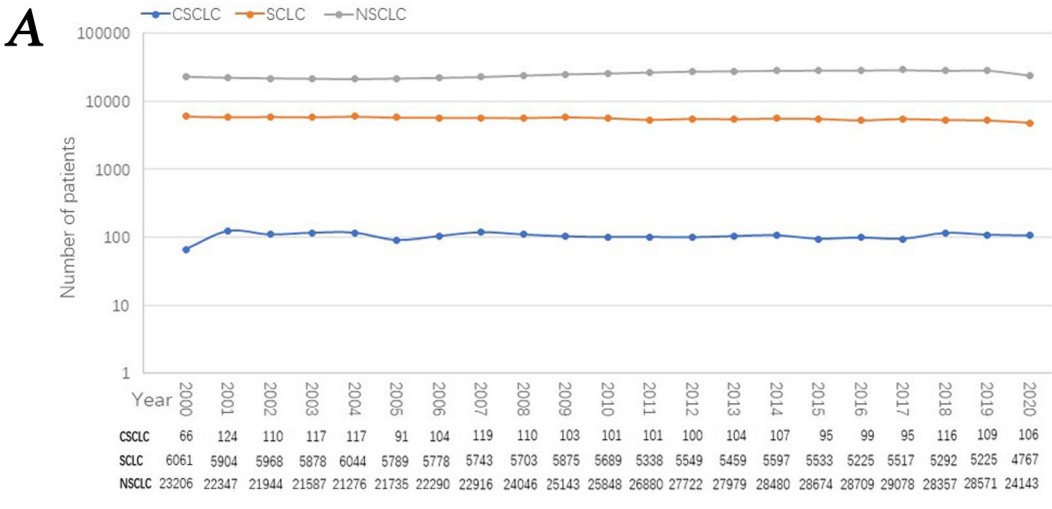

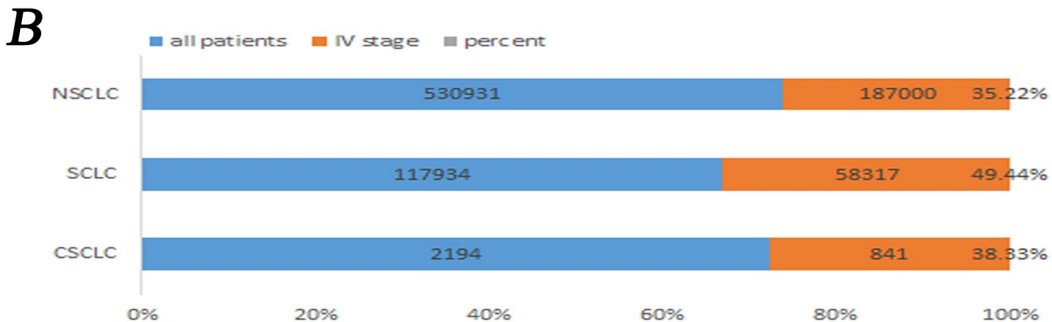

**Fig 3. Epidemiology for CSCLC, SCLC and NSCLC from 2000 to 2020.** A: Semi-logarithmic line chart for number of patients per year in CSCLC, SCLC and NSCLC; B:Total number of patients, IV stage patients and the percentage of stage IV patients in three groups.

Stacked Bar Chart was used to perform the total number of patients, IV stage patients and the percentage of stage IV patients in three groups. The percentage of stage IV patients in SCLC was the highest, followed by CSCLC, and the number of patients in NSCLC was the lowest (Fig 3B).

## Basic characteristics of patients for IV CSCLC, IV SCLC and IV NSCLC before PSM

There were stage IV CSCLC (n = 493), stage IV SCLC (n = 35503) and stage IV NSCLC (n = 122807) groups in this study. Demographics and clinicopathologic characteristics of patients are shown in Table 1. Age of $\geq$65 (67.3%), married (50.7%), upper lobe (49.1%), right (53.1%), T4 (44.8%), N2 (44.4%) and N3 (24.5%) were common in IV CSCLC patients. Gender, race, primary site, T stage, N stage, liver metastasis, lung metastasis, surgery, and radiotherapy were significantly different between stage IV CSCLC and stage IV SCLC group ($p < 0.05$). Race, primary site, N stage, bone metastasis, liver metastasis, lung metastasis and chemotherapy were significantly different in stage IV CSCLC and NSCLC groups ($p < 0.05$).

Lung metastasis (25.6% vs 19.9%) was more common and liver metastasis (31.8% vs 47.7%) was less common in IV CSCLC than in SCLC. Liver metastasis (31.8%VS 17.2%) was more common, lung metastasis (25.6% vs 31.4%) and bone metastases (34.3% vs 40.3%) were less common in IV CSCLC than in NSCLC. More IV CSCLC patients chose chemotherapy (64.1% vs 52.1%) than IV NSCLC patients. More IV CSCLC patients chose surgery (5.1% vs 0.9%) and radiotherapy (43.2% vs 37.3%) than IV SCLC patients.

## Basic characteristics of patients for IV CSCLC, IV SCLC and IV NSCLC after PSM

Random 1:4 nearest-neighbor PSM without replacement to balance all baseline covariates between IV CSCLC, IV SCLC and IV NCSCLC. All covariates were subsequently well balanced both in the 1:4 matched cohort of IV CSCLC (n = 493) vs IV SCLC (n = 1954), and the 1:4 matched cohort of IV CSCLC (n = 493) vs IV NSCLC (n = 1968). ($p \geq 0.05$ and SMD<0.1, Tables 2,3 and Fig 2).

## KM analysis for IV CSCLC, IV SCLC and IV NSCLC before PSM

For the cohorts before PSM, the mOS and mCSS of IV CSCLC were both 6.0 months, IV SCLC patients presented a mOS of 5.00 months and a mCSS of 6 months. The mOS and mCSS of NSCLC patients were 6 months and 7 months. There were statistically significant difference in OS ($p = 0.013$,) and CSS ($p = 0.014$) between stage IV CSCLC and stage IV SCLC groups (Fig 4A and 4B), So did the OS ($p<0.001$) and CSS ($p<0.001$) between stage IV CSCLC and stage IV NSCLC groups (Fig 4C and 4D).

## KM analysis for IV CSCLC, IV SCLC and IV NSCLC after PSM

After PSM, the mOS and mCSS for patients with IV CSCLC were both 6.00 months, while those for SCLC patients were 5.00 months, there were marginally non-significant differences in OS (P = 0.730; Fig 5A) and CSS (P = 0.685; Fig 5B) between IV CSCLC and IV SCLC groups.

In contrast, the mOS and mCSS for IV NSCLC patients remained unchanged at 6 months and 7 months after PSM, respectively. Notably, the mCSS of IV NSCLC patients was longer than that of IV CSCLC patients (7 months versus 6 months). Statistically significant differences

**Table 1. Demographics and clinicopathologic characteristics of patients with CSCLC, SCLC and NSCLC before PSM.**

| Characteristics | CSCLC, N = 493[1] | SCLC, N = 35503[1] | p-value[2] | NSCLC, N = 122807[1] | p-value |
|---|---|---|---|---|---|
| **Age.years.** | | | 0.065 | | 0.381 |
| <65 | 161 (32.7%) | 13,024 (36.7%) | | 42542 (34.6%) | |
| ≥65 | 332 (67.3%) | 22,479 (63.3%) | | 80265 (65.4%) | |
| **Gender** | | | **0.003** | | 0.057 |
| male | 289 (58.6%) | 18,409 (51.9%) | | 66604 (54.2%) | |
| female | 204 (41.4%) | 17,094 (48.1%) | | 56203 (45.8%) | |
| **Race** | | | **0.009** | | **0.002** |
| Black | 57 (11.6%) | 2,981 (8.4%) | | 14930 (12.2%) | |
| White | 405 (82.2%) | 30,976 (87.2%) | | 95575 (77.8%) | |
| Asian or Pacific Islander | 26 (5.3%) | 1,288 (3.6%) | | 11610 (9.5%) | |
| American Indian/Alaska Native | 5 (1.0%) | 258 (0.7%) | | 692 (0.6%) | |
| **Married.status** | | | 0.472 | | 0.256 |
| Married | 250 (50.7%) | 17,082 (48.1%) | | 61977 (50.5) | |
| Divorced | 70 (14.2%) | 5,070 (14.3%) | | 14754 (12.0) | |
| Others | 173 (35.1%) | 13,351 (37.6%) | | 46076 (37.5) | |
| **Primary.Site** | | | **0.005** | | **0.010** |
| Main bronchus | 36 (7.3%) | 3,916 (11.0%) | | 5158 (4.2%) | |
| Upper lobe | 242 (49.1%) | 15,480 (43.6%) | | 59390 (48.4%) | |
| Middle lobe | 14 (2.8%) | 1,249 (3.5%) | | 4940 (4.0%) | |
| Lower lobe | 112 (22.7%) | 7,112 (20.0%) | | 31467 (35.6%) | |
| Others | 89 (18.1[1]) | 7,746 (21.8%) | | 21852 (17.8%) | |
| **Laterality** | | | 0.831 | | 0.655 |
| Left | 196 (39.8%) | 14,100 (39.7%) | | 47460 (38.6%) | |
| Right | 262 (53.1%) | 18,625 (52.5%) | | 67296 (54.8%) | |
| Others | 35 (7.1%) | 2,778 (7.8%) | | 8105 (6.6%) | |
| **T.stage** | | | **0.011** | | 0.254 |
| T0 | 5 (1.0%) | 461 (1.3%) | | 1101 (0.9%) | |
| T1 | 41 (8.3%) | 3,200 (9.0%) | | 13771 (11.2%) | |
| T2 | 121 (24.5%) | 7,204 (20.3%) | | 28899 (23.5%) | |
| T3 | 55 (11.2%) | 3,320 (9.4%) | | 13246 (10.8%) | |
| T4 | 221 (44.8%) | 15,924 (44.9%) | | 50519 (41.1%) | |
| TX | 50 (10.1%) | 5,394 (15.2%) | | 15271 (12.5%) | |
| **N.stage** | | | **<0.001** | | **0.010** |
| N0 | 95 (19.3[1]) | 4,379 (12.3%) | | 30248 (24.6) | |
| N1 | 33 (6.7%) | 2,305 (6.5%) | | 9475 (7.7%) | |
| N2 | 219 (44.4%) | 17,845 (50.3%) | | 49169 (40.0%) | |
| N3 | 121 (24.5%) | 8,828 (24.9%) | | 25870 (21.1%) | |
| NX | 25 (5.1%) | 2,146 (6.0%) | | 8045 (6.6%) | |
| **Bone.Metastasis** | | | 0.520 | | **0.008** |
| Yes | 169 (34.3%) | 12,667 (35.7%) | | 49431 (40.3%) | |
| No | 324 (65.7%) | 22,836 (64.3%) | | 73376 (59.7%) | |
| **Brain.Metastasis** | | | 0.082 | | 0.287 |
| Yes | 140 (28.4%) | 8,870 (25.0%) | | 32155 (26.2%) | |
| No | 353 (71.6%) | 26,633 (75.0%) | | 90652 (73.8%) | |
| **Liver.Metastasis** | | | **<0.001** | | **<0.001** |
| Yes | 157 (31.8%) | 16,940 (47.7%) | | 21077 (17.2%) | |
| No | 336 (68.2%) | 18,563 (52.3%) | | 101730 (82.8%) | |

*(Continued)*

**Table 1.** (Continued)

| Characteristics | CSCLC, N = 493[1] | SCLC, N = 35503[1] | p-value[2] | NSCLC, N = 122807[1] | p-value |
|---|---|---|---|---|---|
| **Lung.Metastasis** | | | **0.002** | | **0.006** |
| Yes | 126 (25.6%) | 7,076 (19.9%) | | 38568 (31.4%) | |
| No | 367 (74.4%) | 28,427 (80.1%) | | 84239 (68.6%) | |
| **Surgery** | | | **<0.001** | | 0.068 |
| Yes | 25 (5.1%) | 329 (0.9%) | | 4250 (3.5%) | |
| No | 468 (94.9%) | 35,174 (99.1%) | | 118557 (96.5%) | |
| **Radiotherapy** | | | **0.008** | | 0.887 |
| Yes | 213 (43.2%) | 13,259 (37.3%) | | 52543 (42.8%) | |
| No | 280 (56.8%) | 22,244 (62.7%) | | 70264 (57.2%) | |
| **Chemotherapy** | | | 0.202 | | **<0.001** |
| Yes | 316 (64.1%) | 23,723 (66.8%) | | 64004 (52.1%) | |
| No | 177 (35.9%) | 11,780 (33.2%) | | 58803 (47.9%) | |

[1]n (%)

were observed in both OS (p<0.001) and CSS (p<0.001) when comparing the IV CSCLC and IV NSCLC groups, as illustrated in Fig 5C and 5D.

## Univariable Cox analysis for IV CSCLC, IV SCLC and IV NSCLC

Before PSM, the results of univariate Cox analysis showed that age, race, N stage, bone metastasis, liver metastasis, surgery, radiotherapy and chemotherapy were significantly associated with OS and CSS in three groups (p<0.05, S1 Table, Figs 6 and 7).

After PSM, age, N stage, bone metastasis, liver metastasis, surgery, radiotherapy, and chemotherapy were commonly associated with OS and CSS in three groups (p<0.05). T stage was correlated with the CSS of IV SCLC and IV CSCLC. Race was commonly associated with OS and CSS of IV NSCLC and IV CSCLC (p<0.05, S2 Table, Figs 6 and 8).

## Multivariate Cox analysis for IV CSCLC, IV SCLC and IV NSCLC

The results of independent influencing factors of the three groups were described by forest plot and the correlation of influencing factors among the three groups was described by Venn diagram (Fig 9).

Before PSM, race, primary site, N stage, bone metastasis, liver metastasis, surgery, chemotherapy and radiotherapy were common independent risk/ protective factors for both CSS and OS of the three groups, besides age was also the independent risk factor for OS of the three groups. Gender, married status, T stage, brain metastasis, and lung metastasis were common independent risk/ protective factors for OS and CSS of SCLC and NSCLC(P<0.05, Table 4, Figs 6 and 7).

After PSM, bone and liver metastases emerged as prevalent independent risk factors for both OS and CSS across the three groups. Additionally, age 65 years and above was identified as an independent risk factor for OS in all three groups. Notably, surgery, radiotherapy, and chemotherapy all were independent protective factors for both CSS and OS in the three groups (p<0.05, Table 5, Figs 6 and 8).

## Prognosis of each treatment modality in IV CSCLC patients

PSM was applied to each treatment modality in IV CSCLC patients in this study. All covariates were subsequently well balanced in the 1:4 matched cohort of surgery (n = 25) vs. no surgery

**Table 2. Demographics and clinicopathologic characteristics of patients with CSCLC and SCLC after 1:4 PSM.**

| Characteristics | CSCLC, N = 493 | SCLC, N = 1954 | SMD | p-value |
|---|---|---|---|---|
| **Age.years.** | | | | 0.645 |
| <65 | 161 (32.7) | 617 (31.6%) | 0.025 | |
| ≥65 | 332 (67.3) | 1,337 (68.4%) | -0.025 | |
| **Gender** | | | | 0.641 |
| male | 289 (58.6) | 1,168 (59.8%) | -0.025 | |
| female | 204 (41.4) | 786 (40.2%) | 0.025 | |
| **Race** | | | | 0.844 |
| Black | 57 (11.6) | 222 (11.4%) | | |
| White | 405 (82.2) | 1,626 (83.2%) | 0.008 | |
| Asian or Pacific Islander | 25 (5.1) | 85 (4.4%) | -0.029 | |
| American Indian/Alaska Native | 6 (1.2) | 21 (1.1%) | -0.005 | |
| **Married.status** | | | | 0.682 |
| Married | 250 (50.7) | 1,033 (52.9%) | -0.047 | |
| Divorced | 70 (14.2) | 260 (13.3%) | 0.024 | |
| Others | 173 (35.1) | 661 (33.8%) | 0.031 | |
| **Primary.Site** | | | | 0.758 |
| Main bronchus | 35 (7.1) | 138 (7.1%) | 0.010 | |
| Upper lobe | 242 (49.1) | 991 (50.7%) | -0.037 | |
| Middle lobe | 14 (2.8) | 59 (3.0%) | -0.010 | |
| Lower lobe | 112 (22.7) | 392 (20.1%) | 0.063 | |
| Others | 90 (18.3) | 374 (19.1%) | -0.024 | |
| **Laterality** | | | | 0.461 |
| Left | 196 (39.8) | 771 (39.5%) | 0.001 | |
| Right | 261 (52.9) | 1,072 (54.9%) | -0.030 | |
| Others | 36 (7.3) | 111 (5.7%) | 0.057 | |
| **T.stage** | | | | 0.955 |
| T0 | 5 (1.0) | 17 (0.9%) | 0.015 | |
| T1 | 41 (8.3) | 144 (7.4%) | 0.035 | |
| T2 | 121 (24.5) | 461 (23.6%) | 0.017 | |
| T3 | 55 (11.2) | 226 (11.6%) | -0.013 | |
| T4 | 221 (44.8) | 902 (46.2%) | -0.025 | |
| TX | 50 (10.1) | 204 (10.4%) | -0.007 | |
| **N.stage** | | | | 0.765 |
| N0 | 95 (19.3) | 374 (19.1%) | -0.011 | |
| N1 | 33 (6.7) | 113 (5.8%) | 0.037 | |
| N2 | 219 (44.4) | 914 (46.8%) | -0.040 | |
| N3 | 121 (24.5) | 443 (22.7%) | 0.046 | |
| NX | 25 (5.1) | 110 (5.6%) | -0.024 | |
| **Bone.Metastasis** | | | | 0.980 |
| Yes | 169 (34.3) | 671 (34.3%) | 0.002 | |
| No | 324 (65.7) | 1,283 (65.7%) | -0.002 | |
| **Brain.Metastasis** | | | | 0.786 |
| Yes | 140 (28.4) | 567 (29.0%) | -0.016 | |
| No | 353 (71.6) | 1,387 (71.0%) | 0.016 | |
| **Liver.Metastasis** | | | | 0.672 |
| Yes | 157 (31.8) | 603 (30.9%) | 0.027 | |
| No | 336 (68.2) | 1,351 (69.1%) | -0.027 | |

*(Continued)*

**Table 2.** (Continued)

| Characteristics | CSCLC, N = 493 | SCLC, N = 1954 | SMD | p-value |
|---|---|---|---|---|
| **Lung.Metastasis** | | | | 0.502 |
| Yes | 126 (25.6) | 471 (24.1%) | 0.030 | |
| No | 367 (74.4) | 1,483 (75.9%) | -0.030 | |
| **Surgery** | | | | 0.426 |
| Yes | 25 (5.1) | 83 (4.2%) | -0.002 | |
| No | 468 (94.9) | 1,871 (95.8%) | 0.002 | |
| **Radiotherapy** | | | | 0.641 |
| Yes | 213 (43.2) | 867 (44.4%) | -0.026 | |
| No | 280 (56.8) | 1,087 (55.6%) | 0.026 | |
| **Chemotherapy** | | | | 0.233 |
| Yes | 316 (64.1) | 1,308 (66.9%) | -0.059 | |
| No | 177 (35.9) | 646 (33.1%) | 0.059 | |

[1]n (%)

groups (n = 95), the 1:1 matched cohort of radiotherapy (n = 129) vs no radiotherapy (n = 129) and chemotherapy (n = 157) vs no chemotherapy (n = 157). (p≥0.05 and SMD<0.1, S3–S5 Tables, S1 Fig).

Before PSM, the mOS for patients undergoing surgery, radiotherapy, and chemotherapy groups was 10.0, 7.0, and 8.0.0 months, surpassing that of patients in the groups who did not receive surgery, radiotherapy, or chemotherapy groups (7.0, 5.0, 1.0 months). Similarly, the mCSS for these treatment groups was 10.0, 7.0, and 9.0 months, longer than that in other three groups (8.0, 6.0, 2.0 months) respectively. KM analysis demonstrated that receiving treatment (surgery, radiotherapy, or chemotherapy) significantly improved survival probabilities in patients with IV CSCLC (p<0.05, S2 Fig).

After PSM, the mOS and mCSS remained comparable among the surgery vs no surgery, radiotherapy vs no radiotherapy, and chemotherapy vs no chemotherapy groups, mirroring the trends observed before PSM. Additionally, KM analysis reconfirmed that undergoing surgery, radiotherapy and chemotherapy treatments all could enhance survival probabilities in IV CSCLC patients (P<0.05, Fig 10).

## Evaluation of different treatment modalities of IV CSCLC, IV SCLC and IV NSCLC

To identify the effect of treatment modalities on OS and CSS for three groups, patients were divided into eight groups according to treatment modalities before PSM: Control: patients were not treated with radiotherapy, chemotherapy or surgery since being diagnosed. Surgery: patients were treated with surgery alone. Chemotherapy: patients were treated with chemotherapy alone. Radiotherapy: patients were treated with radiotherapy alone. Chemoradiotherapy: patients were both treated with radiotherapy and chemotherapy. Surgery + chemotherapy: patients were treated with surgery and chemotherapy. Surgery + radiotherapy: patients were treated with surgery and radiotherapy (data missing in CSCLC group). Surgery + chemoradiotherapy: patients were treated with surgery, chemotherapy and radiotherapy. Characteristics of IV CSCLC, IV SCLC, IV NSCLC were shown in S6–S8 Tables.

Cox regression models were used to adjust potential confounders that were not fully balanced (S9–S11 Tables). The survival curve for three groups was shown in Fig 11 and the

**Table 3. Demographics and clinicopathologic characteristics of patients with CSCLC and NSCLC after 1:4 PSM.**

| Characteristics | CSCLC, N = 493 | NSCLC, N = 1968 | SMD | p-value |
|---|---|---|---|---|
| **Age.years.** | | | | 0.686 |
| <65 | 161 (32.7) | 624 (31.7%) | 0.021 | |
| ≥65 | 332 (67.3) | 1,344 (68.3%) | -0.021 | |
| **Gender** | | | | 0.646 |
| male | 289 (58.6) | 1,176 (59.8%) | -0.025 | |
| female | 204 (41.4) | 792 (40.2%) | 0.025 | |
| **Race** | | | | 0.118 |
| Black | 57 (11.6) | 178 (9.0%) | 0.079 | |
| White | 405 (82.2) | 1,696 (86.2%) | -0.105 | |
| Asian or Pacific Islander | 25 (5.1) | 80 (4.1%) | 0.046 | |
| American Indian/Alaska Native | 6 (1.2) | 14 (0.7%) | 0.046 | |
| **Married.status** | | | | 0.728 |
| Married | 250 (50.7) | 1,008 (51.2%) | -0.008 | |
| Divorced | 70 (14.2) | 301 (15.3%) | -0.036 | |
| Others | 173 (35.1) | 659 (33.5%) | 0.035 | |
| **Primary.Site** | | | | 0.215 |
| Main bronchus | 35 (7.1) | 93 (4.7%) | 0.093 | |
| Upper lobe | 242 (49.1) | 1,025 (52.1%) | -0.062 | |
| Middle lobe | 14 (2.8) | 58 (2.9%) | -0.006 | |
| Lower lobe | 112 (22.7) | 412 (20.9%) | 0.044 | |
| Others | 90 (18.3) | 380 (19.3%) | -0.026 | |
| **Laterality** | | | | 0.745 |
| Left | 196 (39.8) | 752 (38.2%) | 0.029 | |
| Right | 261 (52.9) | 1,057 (53.7%) | -0.014 | |
| Others | 36 (7.3) | 159 (8.1%) | -0.029 | |
| **T.stage** | | | | 0.999 |
| T0 | 5 (1.0) | 19 (1.0%) | 0.005 | |
| T1 | 41 (8.3) | 167 (8.5%) | -0.006 | |
| T2 | 121 (24.5) | 484 (24.6%) | -0.004 | |
| T3 | 55 (11.2) | 226 (11.5%) | -0.010 | |
| T4 | 221 (44.8) | 864 (43.9%) | 0.020 | |
| TX | 50 (10.1) | 208 (10.6%) | -0.013 | |
| **N.stage** | | | | 0.774 |
| N0 | 95 (19.3) | 404 (20.5%) | -0.031 | |
| N1 | 33 (6.7) | 136 (6.9%) | -0.008 | |
| N2 | 219 (44.4) | 897 (45.6%) | -0.026 | |
| N3 | 121 (24.5) | 429 (21.8%) | 0.065 | |
| NX | 25 (5.1) | 102 (5.2%) | -0.005 | |
| **Bone.Metastasis** | | | | 0.554 |
| Yes | 169 (34.3) | 647 (32.9%) | 0.031 | |
| No | 324 (65.7) | 1,321 (67.1%) | -0.031 | |
| **Brain.Metastasis** | | | | 0.438 |
| Yes | 140 (28.4) | 594 (30.2%) | -0.042 | |
| No | 353 (71.6) | 1,374 (69.8%) | 0.042 | |
| **Liver.Metastasis** | | | | 0.515 |
| Yes | 157 (31.8) | 597 (30.3%) | 0.029 | |
| No | 336 (68.2) | 1,371 (69.7%) | -0.029 | |

(*Continued*)

**Table 3.** (Continued)

| Characteristics | CSCLC, N = 493 | NSCLC, N = 1968 | SMD | p-value |
|---|---|---|---|---|
| **Lung.Metastasis** | | | | 0.945 |
| Yes | 126 (25.6) | 500 (25.4%) | 0.005 | |
| No | 367 (74.4) | 1,468 (74.6%) | -0.005 | |
| **Surgery** | | | | 0.535 |
| Yes | 25 (5.1) | 114 (5.8%) | -0.034 | |
| No | 468 (94.9) | 1,854 (94.2%) | 0.034 | |
| **Radiotherapy** | | | | 0.891 |
| Yes | 213 (43.2) | 857 (43.5%) | -0.008 | |
| No | 280 (56.8) | 1,111 (56.5%) | 0.008 | |
| **Chemotherapy** | | | | 0.857 |
| Yes | 316 (64.1) | 1,270 (64.5%) | -0.011 | |
| No | 177 (35.9) | 698 (35.5%) | 0.011 | |

[1]n (%)

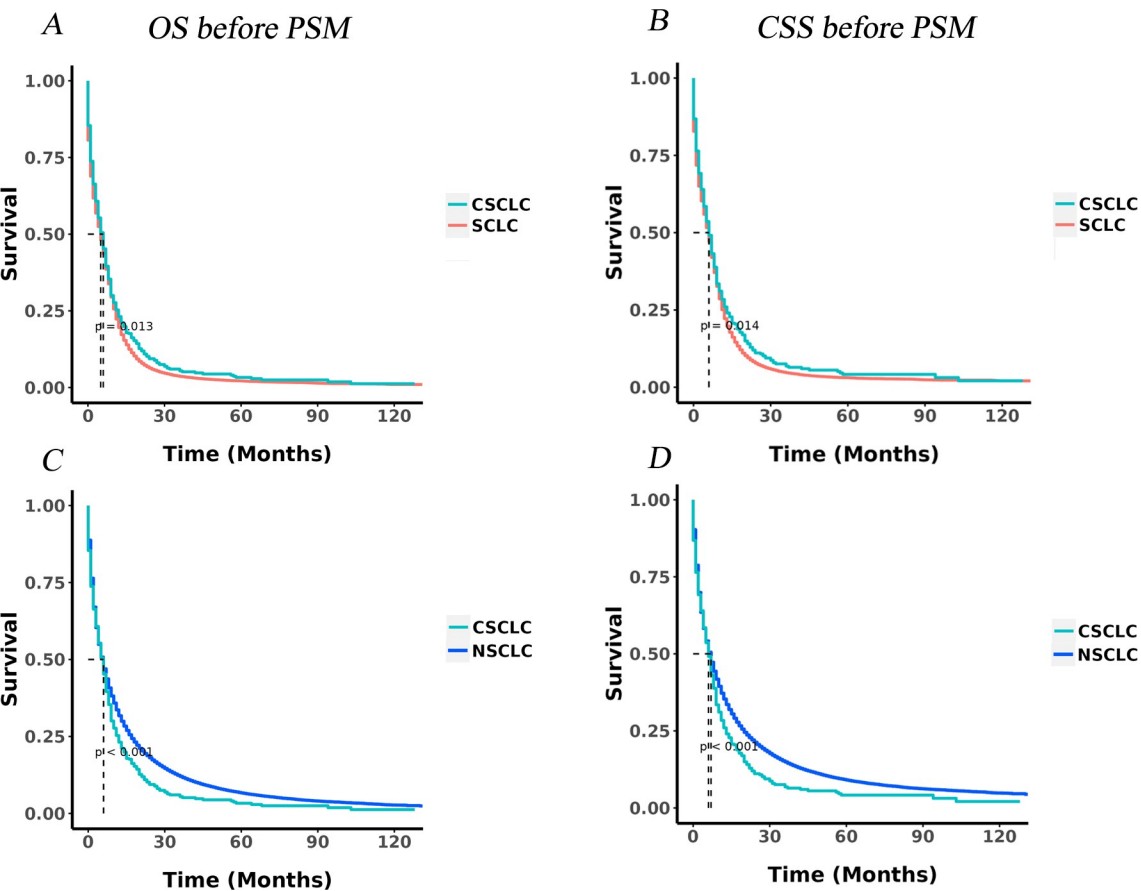

**Fig 4. KM curves in IV CSCLC, IV SCLC and IV NSCLC patients before PSM.** A: The KM curve of OS in CSCLC and SCLC (p = 0.013). B: The KM curve of CSS in CSCLC and SCLC (p = 0.014). C: The KM curve of OS in CSCLC and NSCLC (p<0.001). D: The KM curve of CSS in CSCLC and NSCLC (p<0.001).

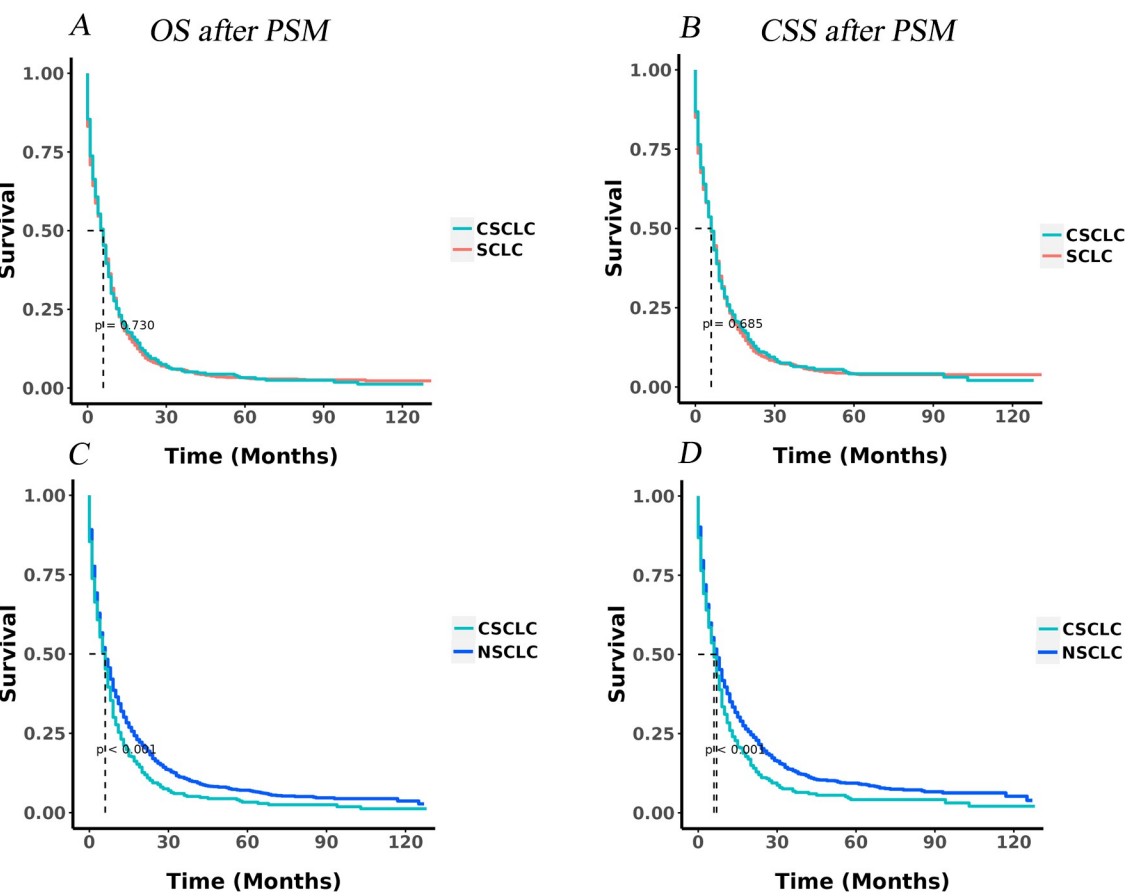

**Fig 5. KM curves in IV CSCLC, IV SCLC and IV NSCLC patients after PSM.** A: The KM curve of OS in CSCLC and SCLC (p = 0.730). B: The KM curve of CSS in CSCLC and SCLC before PSM (p = 0.685). C: The KM curve of OS in CSCLC and NSCLC (p<0.001). D: The KM curve of CSS in CSCLC and NSCLC (p<0.001).

survival curve after adjusted was shown in S3 Fig. The median survival (95% CI) of different treatment modalities in IV CSCLC, IV SCLC and IV NSCLC were shown in Tables 6–8.

The surgery + chemoradiotherapy group for patients with IV CSCLC demonstrated best OS compared to control treatment group (mOS:19.5 months, log-rank P < 0.001; adjusted hazard ratio [HR], 0.16 [95% CI, 0.07–0.36]; P < 0.001) and the surgery + chemotherapy group exhibited the best CSS (mCSS: 16.0 months, log-rank P < 0.001; adjusted HR, 0.15 [95% CI, 0.07–0.36]). Chemoradiotherapy was also a good choice for some IV CSCLC patients who had already lost the opportunity for surgery at the time of first diagnosis (mOS:8 months, adjusted HR, 0.21 [95% CI, 0.16–0.27]; P < 0.001, mCSS: 9 months, adjusted HR, 0.22 [95% CI, 0.17–0.30]; P<0.001).

The mOS and mCSS of Surgery+ chemoradiotherapy group were 13 months and 14 months in IV SCLC, which had a highest probability of survival compared with control group (OS: log-rank P<0.001; adjusted HR, 0.14 [95% CI, 0.12–0.17]; P<0.001; CSS: log-rank P<0.001; adjusted HR, 0.14 [95% CI, 0.11–0.17]; P<0.001). In the IV NSCLC group, the Surgery+ chemotherapy group had best mOS (29 months)and mCSS (35 months) compared with control group (OS: log-rank P<0.001; adjusted HR, 0.32 [95% CI, 0.30–0.35]; P<0.001; CSS: log-rank P<0.001; adjusted HR, 0.33 [95% CI, 0.31–0.36]; P<0.001).

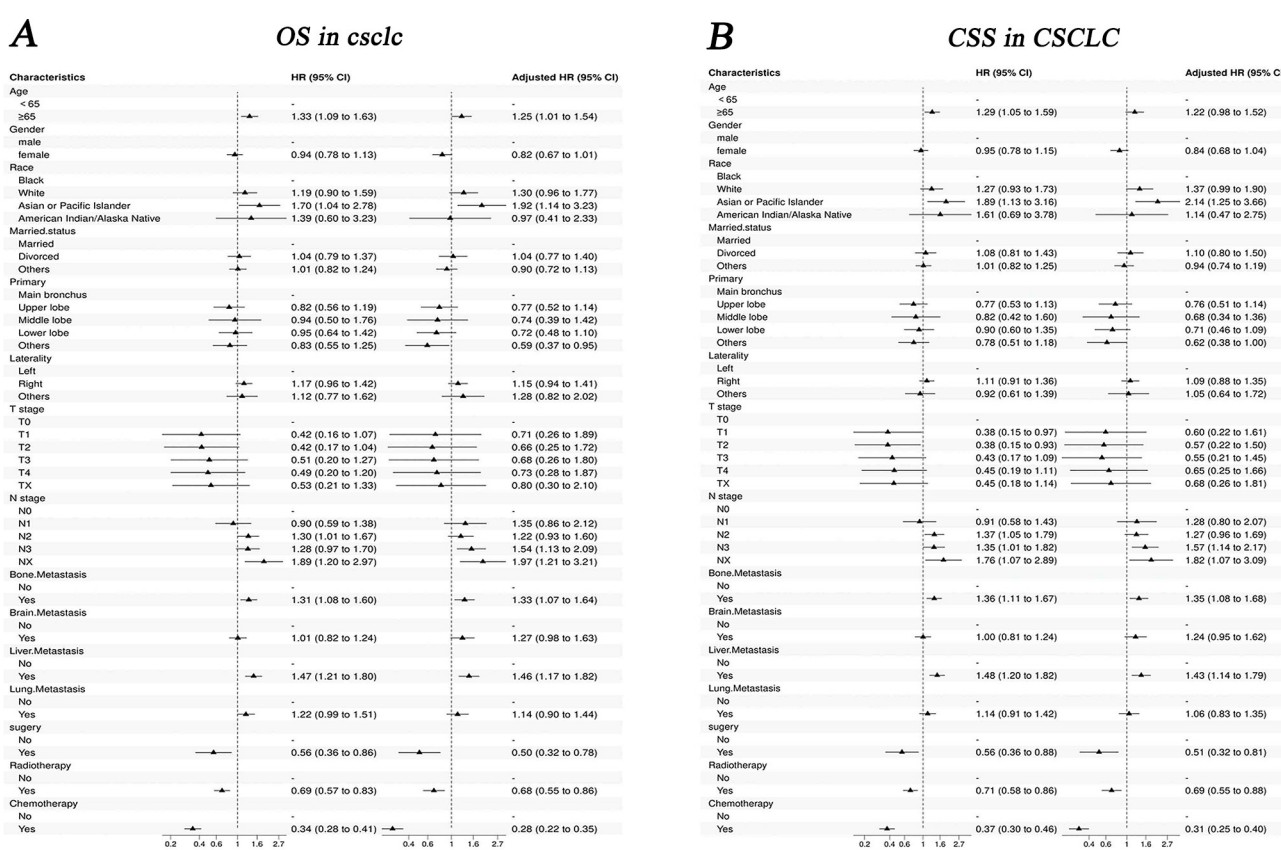

**Fig 6. Forest plot the univariable and multivariate Cox analysis of IV CSCLC.** A:OS in CSCLC; B: CSS in CSCLC.

## Discussion

CSCLC is a rare histological type of lung cancer. In this study, we retrieved the data of CSCLC patients from the SEER database in the past two decades (2194 cases), accounting for 2.2% of the total number of lung cancer patients (100,753 cases). The number of annual incidences has been maintained at around 100 cases, which is consistent with the incidence rate of CSCLC in previous studies ranging from 1% to 14%. The highest detection rate of 14.3% was reported by Fushimi et al, the study found that the detection rate by autopsy was significantly higher than that by biopsy or other cytological methods [15], the main detection method in the SEER database is not autopsy, so the detection rate may be low. The proportion of IV CSCLC patients is 38.33%, which is between that of IV NSCLC (35.22%) and IV SCLC (49.44%). Since some studies have confirmed that CSCLC has histological features in common with NSCLC and SCLC [3, 6–8, 16], it is necessary to further explore the differences in clinical features, prognostic factors, and treatment methods among the three groups.

Our study suggests that the most advanced CSCLC cases are elderly patients, with a higher incidence in the upper lobe and right lung. There are significant differences in clinical characteristics between advanced CSCLC, NSCLC, and SCLC, which are reflected in demographic and tumor features. The probability of liver and lung metastasis in IV CSCLC is between that of IV NSCLC and IV SCLC. This may be due to better treatment outcomes in IV CSCLC, as CSCLC patients are more willing to undergo surgery and chemoradiotherapy compared to the other two groups. However, despite there being no significant difference in survival time (OS and CSS) between IV CSCLC and IV SCLC after PSM, the Kaplan-Meier curve trend and

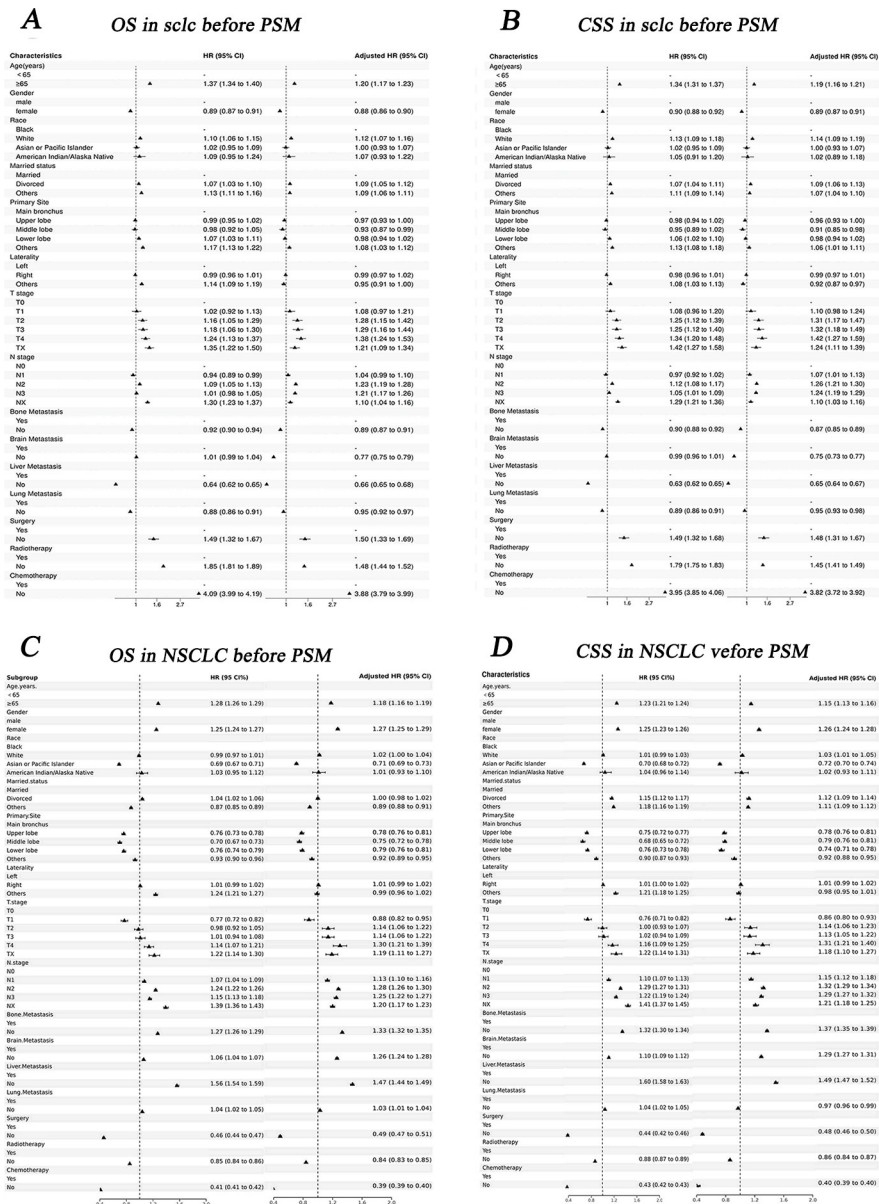

**Fig 7. Forest plot the univariable and multivariate Cox analysis of IV SCLC and IV NSCLC before PSM.** A, B: OS and CSS in SCLC; C, D: OS and CSS in NSCLC.

before PSM results suggest that the survival of IV CSCLC is slightly better than that of IV SCLC. Additionally, both before and after PSM, the survival of IV CSCLC is shorter than that of IV NSCLC (P<0.05), which is consistent with previous studies of CSCLC in I-IV stages [5, 17]. For this situation, SCLC is considered to have the worst prognosis and the highest degree of malignancy among all lung cancer tissue types, especially in the advanced stage [18], it has also been proposed that CSCLC is more closely related to SCLC than to NSCLC [19], the SCLC component is a negative prognostic factor in CSCLC, and there may be a correlation between the proportion of SCLC component and prognosis [20], due to CSCLC is mixed with some components of NSCLC, the survival time will be in the middle of the SCLC and NSCLC.

Subsequently, univariate and multivariate COX analyses were performed on the three groups. Bone metastasis and liver metastasis were identified as independent risk factors for

**Fig 8. Forest plot the univariable and multivariate Cox analysis of IV SCLC and IV NSCLC after PSM.** A, B: OS and CSS in SCLC; C, D: OS and CSS in NSCLC.

prognosis in all three groups. However, brain and lung metastasis were identified as independent risk factors for SCLC and NSCLC, but not for CSCLC. In the study by Zhang et al. [10], it was believed that liver and lung metastases were not prognostic factors for CSCLC patients. However, that study included patients with CSCLC in stages I-IV, so it can be speculated that liver metastasis is only an independent risk factor in advanced CSCLC patients. For SCLC and NSCLC, liver, brain, bone, and lung metastases are all independent risk factors. Previous studies have suggested that SCLC patients with liver metastases had the unfavorable prognosis [21], In our study, liver metastasis in IV CSCLC may also indicate poor prognosis, we found that the probability of liver metastasis in IV CSCLC was higher than that in IV NSCLC (31.8%

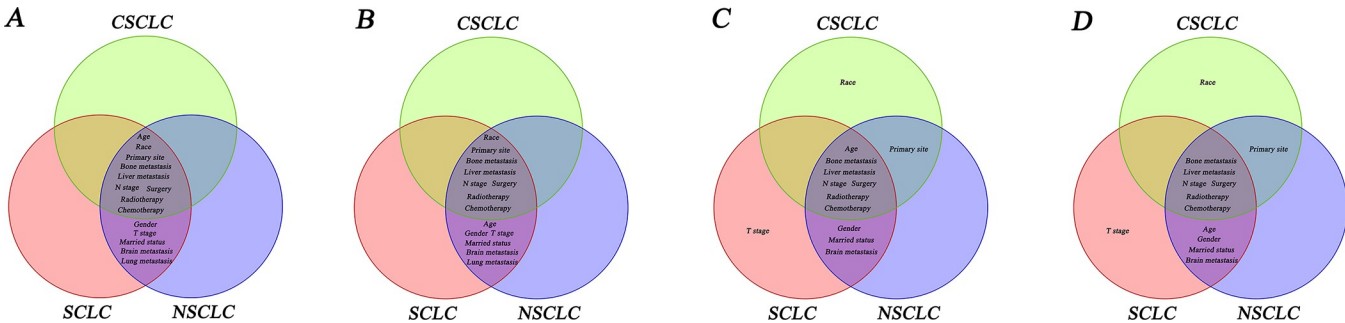

**Fig 9. Venn diagram for correlation of independent influencing factors among IV CSCLC, IV SCLC and IV NSCLC.** A, B: correlation of independent influencing factors among three groups in OS and CSS before PSM; C, D: correlation of independent influencing factors among three groups in OS and CSS after PSM.

vs 17.2%) and the prognosis was significantly poorer than that in IV NSCLC. Bone metastasis is also an important risk factor for the prognosis of IV CSCLC, which can affect CSS and OS of IV CSCLC. Therefore, our study suggests that liver and bone metastasis in patients with advanced CSCLC may indicate poor prognosis. Yang L et al. reported the pivotal role of radiotherapy, chemotherapy, and surgery were critical protective factors for CSCLC, SCLC and NSCLC [22], we also found these treatments were independent protective factors across advanced CSCLC, SCLC and NSCLC in our study either before or after PSM. It is noteworthy that the prognosis of IV CSCLC is also influenced by factors such as age, the primary site and the N stage. Consequently, for individuals diagnosed with advanced CSCLC, it is imperative to accord particular attention to these variables, in order to inform and optimize treatment strategies.

As all three treatment methods were independent protective factors for CSS and OS in CSCLC, we performed separate KM analyses for each treatment method. Patients with IV CSCLC who undergo non-surgical treatment as their initial treatment tend to experience poorer prognoses in clinical practice. In contrast, patients who receive surgical intervention likely represent a subgroup with more localized metastatic disease or favorable staging. Consequently, we introduce PSM to mitigate the impact of selection bias among the various treatment groups, thereby enabling a more accurate assessment of the true effects of different therapeutic strategies. Our findings indicate that all three treatment approaches significantly extend survival time of advanced CSCLC, with statistical significance observed both before and after PSM. Notably, Among the three treatment methods, surgery emerges as the most effective treatment, yielding the longest survival time, closely followed by chemotherapy, aligning with previous research by Men et al, who reported a markedly superior 5-year survival rate among patients with CSCLC who underwent surgical treatment as compared to those who did not [23].

Due to the rare histological type of CSCLC among lung cancers, there is limited real-world data, so there are no detailed guidelines for the treatment of CSCLC [24, 25], especially for advanced CSCLC. As concluded above, all three treatment methods can prolong the survival time of patients with IV CSCLC, but the combined therapy of the three may achieve better results. Thus, we conducted separate KM analysis for each treatment modality in patients with advanced CSCLC, SCLC, and NSCLC. In this study, the combination of surgery and chemoradiotherapy treatment group for patients with IV CSCLC demonstrated superior OS compared to control treatment groups, and the surgery combined chemotherapy treatment group exhibited the best CSS. Thus, Surgery combined with either chemoradiotherapy or chemotherapy may emerge as the preferred treatment modalities for patients diagnosed with advanced

**Table 4. Multivariable Cox analysis of OS and CSS in IV CSCLC, SCLC and NSCLC before PSM.**

| | OS before PSM | | | | | | CSS before PSM | | | | | |
|---|---|---|---|---|---|---|---|---|---|---|---|---|
| | CSCLC, N = 493 | | SCLC, N = 35503 | | NSCLC, N = 122807 | | CSCLC, N = 493 | | SCLC, N = 35503 | | NSCLC, N = 122807 | |
| Characteristic | HR(95%CI) | pvalue | HR(95%CI) | pvalue | HR(95%CI) | pvalue | HR(95%CI) | pvalue | HR(95%CI) | pvalue | HR(95%CI) | pvalue |
| **Age.years.** | | | | | | | | | | | | |
| <65 | | | — | | — | | — | | — | | — | |
| ≥65 | 1.25(1.01, 1.54) | **0.043** | 1.20(1.17, 1.23) | **<0.001** | 1.18 (1.16,1.19) | **<0.001** | 1.22 (0.98,1.52) | 0.079 | 1.19(1.16, 1.21) | **<0.001** | 1.15 (1.13,1.16) | **<0.001** |
| **Gender** | | | | | | | | | | | | |
| male | — | | — | | — | | — | | — | | — | |
| female | 0.82(0.67, 1.01) | 0.060 | 0.88 (0.86,0.90) | **<0.001** | 1.27 (1.25,1.29) | **<0.001** | 0.84(0.68, 1.04) | 0.114 | 0.89(0.87, 0.91) | **<0.001** | 1.26 (1.24,1.28) | **<0.001** |
| **Race** | | | | | | | | | | | | |
| Black | — | | — | | — | | — | | — | | — | |
| White | 1.3(0.96, 1.77) | 0.091 | 1.12 (1.07,1.16) | **<0.001** | 1.02 (1.00,1.04) | 0.066 | 1.37(0.99, 1.90) | 0.056 | 1.14(1.09, 1.19) | **<0.001** | 1.03 (1.01,1.05) | **<0.001** |
| Asian or Pacific Islander | 1.92(1.14, 3.23) | **0.014** | 1.00 (0.93,1.07) | 0.899 | 0.71 (0.69,0.73) | **<0.001** | 2.14(1.25, 3.66) | **0.006** | 1.00(0.93, 1.07) | 0.935 | 0.72 (0.70,0.74) | **<0.001** |
| American Indian/ Alaska Native | 0.97 (0.41,2.33) | 0.952 | 1.07 (0.93,1.22) | 0.342 | 1.01 (0.93,1.10) | 0.805 | 1.14(0.47, 2.75) | 0.778 | 1.02(0.89, 1.18) | 0.779 | 1.02 (0.93,1.11) | 0.734 |
| **Married.status** | | | | | | | | | | | | |
| Married | — | | — | | — | | — | | — | | — | |
| Divorced | 1.04(0.77, 1.40) | 0.817 | 1.09(1.05, 1.12) | **<0.001** | 1.00 (0.98,1.02) | 0.888 | 1.1(0.80, 1.50) | 0.555 | 1.09(1.06, 1.13) | **<0.001** | 1.12 (1.09,1.14) | **<0.001** |
| Others | 0.9(0.72, 1.13) | 0.374 | 1.09(1.06, 1.11) | **<0.001** | 0.89 (0.88,0.91) | **<0.001** | 0.94(0.74, 1.19) | 0.604 | 1.07(1.04, 1.10) | **<0.001** | 1.11 (1.09,1.12) | **<0.001** |
| **Primary.Site** | | | | | | | | | | | | |
| Main bronchus | — | | — | | — | | — | | — | | — | |
| Upper lobe | 0.77(0.52, 1.14) | 0.197 | 0.97(0.93, 1.00) | 0.071 | 0.78 (0.76,0.81) | **<0.001** | 0.76(0.51, 1.14) | 0.179 | 0.96(0.93, 1.00) | **0.043** | 0.78 (0.76,0.81) | **<0.001** |
| Middle lobe | 0.74(0.39, 1.42) | 0.366 | 0.93(0.87, 0.99) | **0.031** | 0.75 (0.72,0.78) | **<0.001** | 0.68(0.34, 1.36) | 0.280 | 0.91(0.85, 0.98) | **0.008** | 0.79 (0.76,0.81) | **<0.001** |
| Lower lobe | 0.72(0.48, 1.10) | 0.132 | 0.98(0.94, 1.02) | 0.315 | 0.79 (0.76,0.81) | **<0.001** | 0.71(0.46, 1.09) | 0.117 | 0.98(0.94, 1.02) | 0.265 | 0.74 (0.71,0.78) | **<0.001** |
| Others | 0.59(0.37, 0.95) | **0.030** | 1.08(1.03, 1.12) | **0.001** | 0.92 (0.89,0.95) | **<0.001** | 0.62(0.38, 1.00) | **0.050** | 1.06(1.01, 1.11) | **0.014** | 0.92 (0.88,0.95) | **<0.001** |
| **Laterality** | | | | | | | | | | | | |
| Left | — | | — | | — | | — | | — | | — | |
| Right | 1.15(0.94, 1.41) | 0.180 | 0.99(0.97, 1.02) | 0.515 | 1.01 (0.99,1.02) | 0.403 | 1.09(0.88, 1.35) | 0.407 | 0.99(0.97, 1.01) | 0.386 | 1.01 (0.99,1.02) | 0.226 |
| Others | 1.28(0.82, 2.02) | 0.279 | 0.95(0.91, 1.00) | 0.071 | 0.99 (0.96,1.02) | 0.359 | 1.05(0.64, 1.72) | 0.853 | 0.92(0.87, 0.97) | **0.002** | 0.98 (0.95,1.01) | 0.129 |
| **T.stage** | | | | | | | | | | | | |
| T0 | — | | — | | — | | — | | — | | — | |
| T1 | 0.71(0.26, 1.89) | 0.491 | 1.08(0.97, 1.21) | 0.15 | 0.88 (0.82,0.95) | **<0.001** | 0.6(0.22, 1.61) | 0.310 | 1.1(0.98, 1.24) | 0.111 | 0.86 (0.80,0.93) | **<0.001** |
| T2 | 0.66(0.25, 1.72) | 0.396 | 1.28(1.15, 1.42) | **<0.001** | 1.14 (1.06,1.22) | **<0.001** | 0.57 (0.22,1.50) | 0.255 | 1.31(1.17, 1.47) | **<0.001** | 1.14 (1.06,1.23) | **<0.001** |
| T3 | 0.68(0.26, 1.80) | 0.441 | 1.29(1.16, 1.44) | **<0.001** | 1.14 (1.06,1.22) | **<0.001** | 0.55(0.21, 1.45) | 0.223 | 1.32(1.18, 1.49) | **<0.001** | 1.13 (1.05,1.22) | **<0.001** |
| T4 | 0.73(0.28, 1.87) | 0.514 | 1.38(1.24, 1.53) | **<0.001** | 1.30 (1.21,1.39) | **<0.001** | 0.65(0.25, 1.66) | 0.367 | 1.42(1.27, 1.59) | **<0.001** | 1.31 (1.21,1.40) | **<0.001** |
| TX | 0.8(0.30, 2.10) | 0.649 | 1.21(1.09, 1.34) | **<0.001** | 1.19 (1.11,1.27) | **<0.001** | 0.68(0.26, 1.81) | 0.441 | 1.24(1.11, 1.39) | **<0.001** | 1.18 (1.10,1.27) | **<0.001** |

(*Continued*)

**Table 4.** (Continued)

| | OS before PSM | | | | | | CSS before PSM | | | | | |
| | CSCLC, N = 493 | | SCLC, N = 35503 | | NSCLC, N = 122807 | | CSCLC, N = 493 | | SCLC, N = 35503 | | NSCLC, N = 122807 | |
| Characteristic | HR(95%CI) | pvalue | HR(95%CI) | pvalue | HR(95%CI) | pvalue | HR(95%CI) | pvalue | HR(95%CI) | pvalue | HR(95%CI) | pvalue |
|---|---|---|---|---|---|---|---|---|---|---|---|---|
| **N.stage** | | | | | | | | | | | | |
| N0 | — | | — | | — | | — | | — | | — | |
| N1 | 1.35(0.86, 2.12) | 0.197 | 1.04(0.99, 1.10) | 0.111 | 1.13 (1.10,1.16) | <0.001 | 1.28(0.80, 2.07) | 0.305 | 1.07(1.01, 1.13) | **0.020** | 1.15 (1.12,1.18) | <0.001 |
| N2 | 1.22(0.93, 1.60) | 0.146 | 1.23(1.19, 1.28) | <0.001 | 1.28 (1.26,1.30) | <0.001 | 1.27(0.96, 1.69) | 0.100 | 1.26(1.21, 1.30) | <0.001 | 1.32 (1.29,1.34) | <0.001 |
| N3 | 1.54(1.13, 2.09) | **0.006** | 1.21(1.17, 1.26) | <0.001 | 1.25 (1.22,1.27) | **0.003** | 1.57(1.14, 2.17) | 0.006 | 1.24(1.19, 1.29) | <0.001 | 1.29 (1.27,1.32) | <0.001 |
| NX | 1.97(1.21, 3.21) | **0.007** | 1.1(1.04, 1.16) | **0.001** | 1.20 (1.17,1.23) | **0.008** | 1.82(1.07, 3.09) | **0.026** | 1.1(1.03, 1.16) | **0.002** | 1.21 (1.18,1.25) | <0.001 |
| **Bone.Metastasis** | | | | | | | | | | | | |
| No | — | | — | | — | | — | | | | — | |
| Yes | 1.33(1.07, 1.64) | **0.009** | 1.13(1.10, 1.15) | <0.001 | 1.33 (1.32,1.35) | <0.001 | 1.35 (1.08,1.68) | **0.008** | 1.15(1.12, 1.18) | <0.001 | 1.37 (1.35,1.39) | <0.001 |
| **Brain.Metastasis** | | | | | | | | | | | | |
| No | — | | — | | — | | — | | — | | — | |
| Yes | 1.27(0.98, 1.63) | 0.070 | 1.3(1.27, 1.34) | <0.001 | 1.26 (1.24,1.28) | <0.001 | 1.24(0.95, 1.62) | 0.107 | 1.34(1.30, 1.38) | <0.001 | 1.29 (1.27,1.31) | <0.001 |
| **Liver.Metastasis** | | | | | | | | | | | | |
| No | — | | — | | — | | — | | — | | — | |
| Yes | 1.46(1.17, 1.82) | <0.001 | 1.51(1.48, 1.55) | <0.001 | 1.47 (1.44,1.49) | <0.001 | 1.43(1.14, 1.79) | **0.002** | 1.53(1.49, 1.56) | <0.001 | 1.49 (1.47,1.52) | <0.001 |
| **Lung.Metastasis** | | | | | | | | | | | | |
| No | — | | — | | — | | — | | — | | — | |
| Yes | 1.14(0.90, 1.44) | 0.265 | 1.05(1.03, 1.08) | <0.001 | 1.03 (1.01,1.04) | <0.001 | 1.06(0.83, 1.35) | 0.642 | 1.05(1.02, 1.08) | **0.001** | 0.97 (0.96,0.99) | <0.001 |
| **Surgery** | | | | | | | | | | | | |
| No | — | | — | | — | | — | | — | | — | |
| Yes | 0.5(0.32, 0.78) | **0.002** | 0.67(0.59, 0.75) | <0.001 | 0.49 (0.47,0.51) | <0.001 | 0.51(0.32, 0.81) | **0.005** | 0.68(0.60, 0.76) | <0.001 | 0.48 (0.46,0.50) | <0.001 |
| **Radiotherapy** | | | | | | | | | | | | |
| No | — | | — | | — | | — | | — | | — | |
| Yes | 0.68(0.55, 0.86) | **0.001** | 0.68(0.66, 0.69) | <0.001 | 0.84 (0.83,0.85) | <0.001 | 0.69 (0.55,0.88) | **0.002** | 0.69(0.67, 0.71) | <0.001 | 0.86 (0.84,0.87) | <0.001 |
| **Chemotherapy** | | | | | | | | | | | | |
| No | — | | — | | — | | —— | | — | | — | |
| Yes | 0.28(0.22, 0.35) | <0.001 | 0.26(0.25, 0.26) | <0.001 | 0.39 (0.39,0.40) | <0.001 | 0.31(0.25, 0.40) | <0.001 | 0.26 (0.26,0.27) | <0.001 | 0.40 (0.39,0.40) | <0.001 |

CSCLC, consistent with study of He J etal.: trimodality therapy could improve CSCLC survival outcomes compared to other therapies [20]. However, for some advanced patients who have already lost the opportunity for surgery at the time of first diagnosis, our study suggests that chemoradiotherapy is also a good choice, with the median survival time second only to surgery combined treatment. Similarly, trimodality therapy is the most effective therapy for patients with advanced SCLC, but surgery combined chemotherapy treatment is the most effective therapy for patients with advanced NSCLC.

**Table 5. Multivariable Cox analysis of OS and CSS in IV CSCLC, SCLC and NSCLC after 1:4 PSM.**

| | OS after PSM | | | | | | CSS after PSM | | | | | |
|---|---|---|---|---|---|---|---|---|---|---|---|---|
| | CSCLC, N = 493 | | SCLC, N = 1954 | | NSCLC, N = 1968 | | CSCLC, N = 493 | | SCLC, N = 1954 | | NSCLC, N = 1968 | |
| Characteristic | HR(95CI) | pvalue | HR(95CI) | pvalue | HR(95CI) | pvalue | HR(95Cl) | pvalue | HR(95CI) | pvalue | HR(95CI) | pvalue |
| **Age.years.** | | | | | | | | | | | | |
| <65 | — | | — | | — | | — | | — | | — | |
| ≥65 | 1.25(1.01, 1.54) | **0.043** | 1.28(1.15, 1.42) | **<0.001** | 1.32(1.19, 1.47) | **<0.001** | 1.22 (0.98,1.52) | 0.079 | 1.26(1.13, 1.41) | **<0.001** | 1.28(1.15, 1.44) | **<0.001** |
| **Gender** | | | | | | | | | | | — | |
| male | — | | — | | — | | — | | — | | — | |
| female | 0.82(0.67, 1.01) | 0.060 | 0.86(0.78, 0.95) | **0.003** | 0.78(0.71, 0.87) | **<0.001** | 0.84(0.68, 1.04) | 0.114 | 0.86(0.77, 0.95) | **0.005** | 0.79(0.71, 0.88) | **<0.001** |
| **Race** | | | | | | | | | | | — | |
| Black | — | | — | | — | | — | | — | | — | |
| White | 1.3(0.96, 1.77) | 0.091 | 1.16(1.00, 1.36) | 0.057 | 1.1(0.93, 1.31) | 0.28 | 1.37(0.99, 1.90) | 0.056 | 1.17(0.99, 1.37) | 0.058 | 1.13(0.94, 1.35) | 0.202 |
| Asian or Pacific Islander | 1.92(1.14, 3.23) | **0.014** | 1.02(0.78, 1.35) | 0.868 | 0.78(0.58, 1.06) | 0.118 | 2.14(1.25, 3.66) | **0.006** | 0.98(0.74, 1.31) | 0.908 | 0.8(0.58, 1.10) | 0.171 |
| American Indian/Alaska Native | 0.97 (0.41,2.33) | 0.952 | 1.07(0.67, 1.72) | 0.77 | 1.75(0.99, 3.11) | 0.056 | 1.14(0.47, 2.75) | 0.778 | 0.9(0.54, 1.52) | 0.703 | 1.79(0.98, 3.26) | 0.056 |
| **Married.status** | | | | | | | | | | | — | |
| Married | — | | — | | — | | — | | — | | — | |
| Divorced | 1.04(0.77, 1.40) | 0.817 | 1.23(1.07, 1.43) | **0.005** | 1.24(1.08, 1.43) | **0.003** | 1.1(0.80, 1.50) | 0.555 | 1.24(1.07, 1.45) | **0.004** | 1.2(1.04, 1.40) | **0.015** |
| Others | 0.9(0.72, 1.13) | 0.374 | 1.19(1.06, 1.32) | **0.002** | 1.16(1.04, 1.30) | **0.010** | 0.94(0.74, 1.19) | 0.604 | 1.16(1.04, 1.31) | **0.010** | 1.14(1.01, 1.28) | **0.033** |
| **Primary.Site** | | | | | | | | | | | — | |
| Main bronchus | — | | — | | — | | — | | — | | — | |
| Upper lobe | 0.77(0.52, 1.14) | 0.197 | 1.18(0.97, 1.43) | 0.097 | 0.73(0.58, 0.91) | **0.006** | 0.76(0.51, 1.14) | 0.179 | 1.14(0.93, 1.39) | 0.203 | 0.73(0.57, 0.92) | **0.009** |
| Middle lobe | 0.74(0.39, 1.42) | 0.366 | 1.06(0.76, 1.48) | 0.738 | 0.66(0.46, 0.94) | **0.023** | 0.68(0.34, 1.36) | 0.280 | 1.02(0.72, 1.45) | 0.916 | 0.67(0.46, 0.98) | **0.039** |
| Lower lobe | 0.72(0.48, 1.10) | 0.132 | 1.2(0.97, 1.48) | 0.088 | 0.65(0.51, 0.83) | **<0.001** | 0.71(0.46, 1.09) | 0.117 | 1.16(0.93, 1.44) | 0.188 | 0.65(0.50, 0.84) | **0.001** |
| Others | 0.59(0.37, 0.95) | **0.030** | 1.17(0.94, 1.46) | 0.164 | 0.77(0.60, 1.00) | **0.05** | 0.62(0.38, 1.00) | **0.050** | 1.13(0.90, 1.42) | 0.283 | 0.78(0.59, 1.02) | 0.069 |
| **Laterality** | | | | | | | | | | | — | |
| Left | — | | — | | — | | — | | — | | — | |
| Right | 1.15(0.94, 1.41) | 0.180 | 1.03(0.93, 1.14) | 0.512 | 0.96(0.87, 1.07) | 0.462 | 1.09(0.88, 1.35) | 0.407 | 1.03(0.93, 1.15) | 0.533 | 0.94(0.84, 1.04) | 0.238 |
| Others | 1.28(0.82, 2.02) | 0.279 | 0.86(0.67, 1.11) | 0.253 | 0.95(0.76, 1.18) | 0.64 | 1.05(0.64, 1.72) | 0.853 | 0.85(0.65, 1.12) | 0.244 | 0.94(0.75, 1.18) | 0.602 |
| **T.stage** | | | | | | | | | | | — | |
| T0 | — | | — | | — | | — | | — | | — | |
| T1 | 0.71(0.26, 1.89) | 0.491 | 1.42(0.76, 2.66) | 0.272 | 0.87(0.48, 1.58) | 0.656 | 0.6(0.22, 1.61) | 0.310 | 1.34(0.70, 2.59) | 0.38 | 0.79(0.44, 1.44) | 0.444 |
| T2 | 0.66(0.25, 1.72) | 0.396 | 1.77(0.96, 3.27) | 0.065 | 1.23(0.69, 2.18) | 0.483 | 0.57 (0.22,1.50) | 0.255 | 1.82(0.96, 3.44) | 0.066 | 1.14(0.64, 2.03) | 0.657 |
| T3 | 0.68(0.26, 1.80) | 0.441 | 1.77(0.95, 3.29) | 0.071 | 1.25(0.70, 2.24) | 0.449 | 0.55(0.21, 1.45) | 0.223 | 1.65(0.86, 3.17) | 0.129 | 1.12(0.62, 2.01) | 0.708 |
| T4 | 0.73(0.28, 1.87) | 0.514 | 1.88(1.03, 3.45) | **0.041** | 1.47(0.83, 2.59) | 0.185 | 0.65(0.25, 1.66) | 0.367 | 1.87(0.99, 3.53) | **0.050** | 1.32(0.75, 2.34) | 0.341 |
| TX | 0.8(0.30, 2.10) | 0.649 | 1.63(0.89, 2.98) | 0.117 | 1.52(0.85, 2.70) | 0.155 | 0.68(0.26, 1.81) | 0.441 | 1.63(0.87, 3.08) | 0.129 | 1.32(0.74, 2.36) | 0.344 |

*(Continued)*

**Table 5.** (Continued)

| | OS after PSM | | | | | | CSS after PSM | | | | | |
| | CSCLC, N = 493 | | SCLC, N = 1954 | | NSCLC, N = 1968 | | CSCLC, N = 493 | | SCLC, N = 1954 | | NSCLC, N = 1968 | |
| Characteristic | HR(95CI) | pvalue | HR(95CI) | pvalue | HR(95CI) | pvalue | HR(95Cl) | pvalue | HR(95CI) | pvalue | HR(95CI) | pvalue |
|---|---|---|---|---|---|---|---|---|---|---|---|---|
| **N.stage** | | | | | | | | | | | — | |
| N0 | — | | — | | — | | — | | — | | — | |
| N1 | 1.35(0.86, 2.12) | 0.197 | 1.15(0.92, 1.44) | 0.21 | 1.07(0.86, 1.33) | 0.526 | 1.28(0.80, 2.07) | 0.305 | 1.14(0.90, 1.44) | 0.268 | 1.12(0.89, 1.41) | 0.329 |
| N2 | 1.22(0.93, 1.60) | 0.146 | 1.23(1.08, 1.41) | **0.002** | 1.2(1.05, 1.37) | **0.007** | 1.27(0.96, 1.69) | 0.100 | 1.21(1.06, 1.40) | **0.007** | 1.27(1.11, 1.46) | **<0.001** |
| N3 | 1.54(1.13, 2.09) | **0.006** | 1.18(1.01, 1.38) | **0.035** | 1.18(1.01, 1.38) | **0.033** | 1.57(1.14, 2.17) | 0.006 | 1.17(1.00, 1.38) | 0.052 | 1.23(1.04, 1.45) | **0.013** |
| NX | 1.97(1.21, 3.21) | **0.007** | 1.14(0.89, 1.45) | 0.308 | 1.24(0.96, 1.59) | 0.100 | 1.82(1.07, 3.09) | **0.026** | 1.01(0.78, 1.31) | 0.935 | 1.32(1.01, 1.72) | **0.042** |
| **Bone.Metastasis** | | | | | | | | | | | — | |
| No | — | | — | | — | | — | | — | | — | |
| Yes | 1.33(1.07, 1.64) | **0.009** | 1.24(1.12, 1.38) | **<0.001** | 1.43(1.28, 1.59) | **<0.001** | 1.35 (1.08,1.68) | **0.008** | 1.27(1.15, 1.42) | **<0.001** | 1.41(1.27, 1.58) | **<0.001** |
| **Brain.Metastasis** | | | | | | | | | | | — | |
| No | — | | — | | — | | — | | — | | — | |
| Yes | 1.27(0.98, 1.63) | 0.070 | 1.32(1.17, 1.48) | **<0.001** | 1.27(1.13, 1.43) | **<0.001** | 1.24(0.95, 1.62) | 0.107 | 1.35(1.20, 1.52) | **<0.001** | 1.33(1.17, 1.51) | **<0.001** |
| **Liver.Metastasis** | | | | | | | | | | | — | |
| No | — | | — | | — | | — | | — | | — | |
| Yes | 1.46(1.17, 1.82) | **<0.001** | 1.61(1.45, 1.80) | **<0.001** | 1.25(1.12, 1.39) | **<0.001** | 1.43(1.14, 1.79) | **0.002** | 1.64(1.46, 1.83) | **<0.001** | 1.26(1.12, 1.41) | **<0.001** |
| **Lung.Metastasis** | | | | | | | | | | | — | |
| No | — | | — | | — | | — | | — | | — | |
| Yes | 1.14(0.90, 1.44) | 0.265 | 1.08(0.97, 1.21) | 0.157 | 1.01(0.90, 1.14) | 0.833 | 1.06(0.83, 1.35) | 0.642 | 1.07(0.95, 1.20) | 0.268 | 1.01(0.89, 1.14) | 0.935 |
| **Surgery** | | | | | | | | | | | — | |
| No | — | | — | | — | | — | | — | | — | |
| Yes | 0.5(0.32, 0.78) | **0.002** | 0.75(0.58, 0.95) | **0.018** | 0.47(0.37, 0.60) | **<0.001** | 0.51(0.32, 0.81) | **0.005** | 0.71(0.55, 0.92) | **0.010** | 0.46(0.35, 0.59) | **<0.001** |
| **Radiotherapy** | | | | | | | | | | | — | |
| No | — | | — | | — | | — | | — | | — | |
| Yes | 0.68(0.55, 0.86) | **0.001** | 0.69(0.62, 0.77) | **<0.001** | 0.91(0.81, 1.02) | **0.099** | 0.69 (0.55,0.88) | **0.002** | 0.7(0.62, 0.78) | **<0.001** | 0.88(0.78, 0.99) | **0.032** |
| **Chemotherapy** | | | | | | | | | | | — | |
| No | — | | — | | — | | — | | — | | — | |
| Yes | 0.28(0.22, 0.35) | **<0.001** | 0.25(0.22, 0.28) | **<0.001** | 0.39(0.35, 0.43) | **<0.001** | 0.31(0.25, 0.40) | **<0.001** | 0.25(0.22, 0.29) | **<0.001** | 0.39(0.35, 0.44) | **<0.001** |

Since CSCLC has components of both SCLC and NSCLC, and SCLC and CSCLC exhibit similarities in terms of their survival probabilities and preferred treatment modalities, can we draw insights from the treatment protocols of NSCLC or SCLC? A study by Luo et al. found that CSCLC patients who received the classic chemotherapy regimen for SCLC (etoposide and cisplatin) presented a survival benefit [26], this study retrospectively compared the efficacy of the NIP (navelbine + ifosfamide + cisplatin) and EP (etoposide + cisplatin) regimens as first-line treatment options for III-IV stage CSCLC, the results suggest that the EP regimen may provide better survival benefits compared to the NIP regimen. However, some studies have mentioned that NSCLC is less sensitive to the EP regimen, and CSCLC contains NSCLC

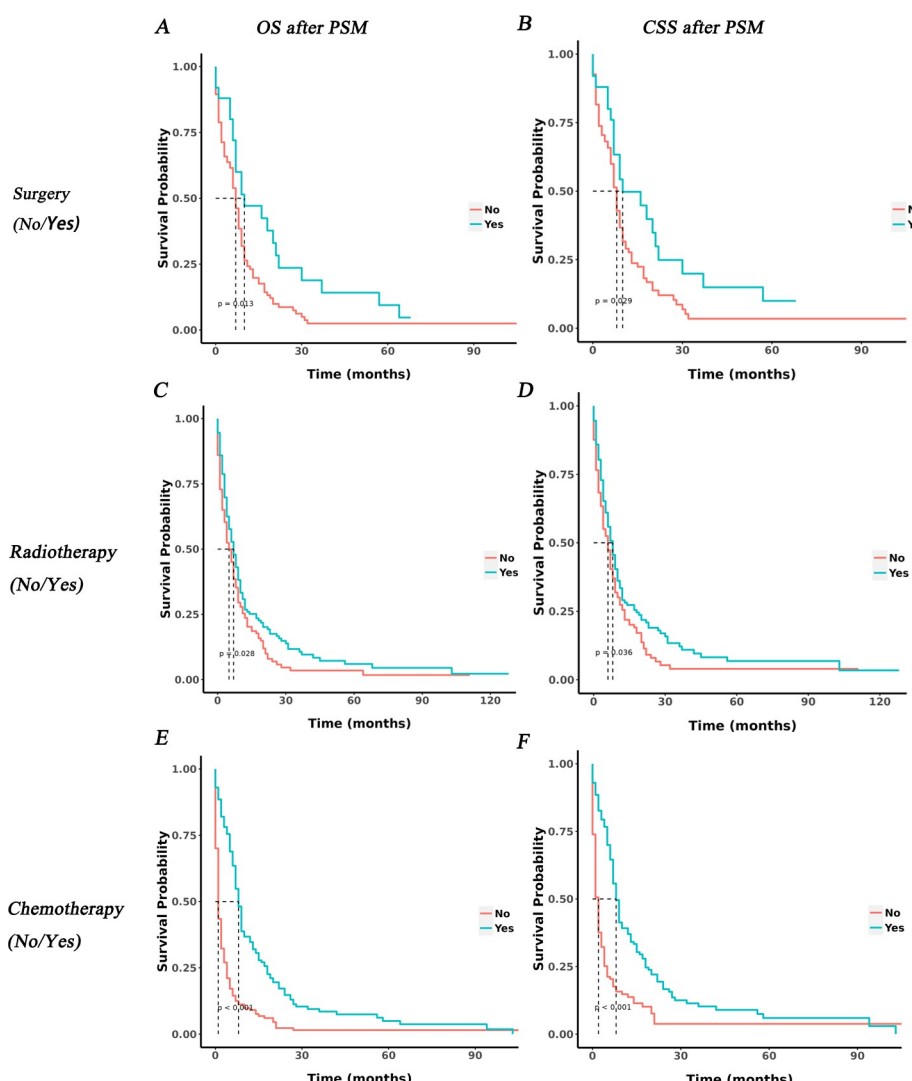

**Fig 10. KM curves of IV CSCLC patients with each treatment modality after PSM.** A, B: KM curve of OS and CSS in surgery and non-surgical treatment groups. C, D: KM curve of OS and CSS in radiotherapy and non-radiotherapy treatment groups. E, F: KM curve of OS and CSS in chemotherapy and non-chemotherapy treatment groups.

components, so the clinical benefits of using the EP regimen for CSCLC may not be as effective as for SCLC [14]. EGFR-TKIs are widely used in NSCLC with EGFR mutations, but there have been no randomized clinical trials to evaluate their efficacy in the treatment of CSCLC, only small case series studies have suggested that EGFR-TKIs may be helpful in the treatment of CSCLC and SCLC, so the efficacy of EGFR-TKIs in the treatment of CSCLC or SCLC may not be as good as in NSCLC [27, 28]. In summary, for patients with advanced CSCLC, if there are indications for surgery, the surgical combination therapy should be given priority. If the opportunity for surgery has been lost, chemoradiotherapy is preferred treatment and the chemotherapy regimen can refer to SCLC.

There are some limitations to our study. Firstly, the SEER database does not provide information on some patient characteristics such as smoking, past medical history, specific chemotherapy and radiotherapy regimens, immunotherapy and targeted therapy, which may affect our results. Secondly, no further molecular level analysis and comparison were performed

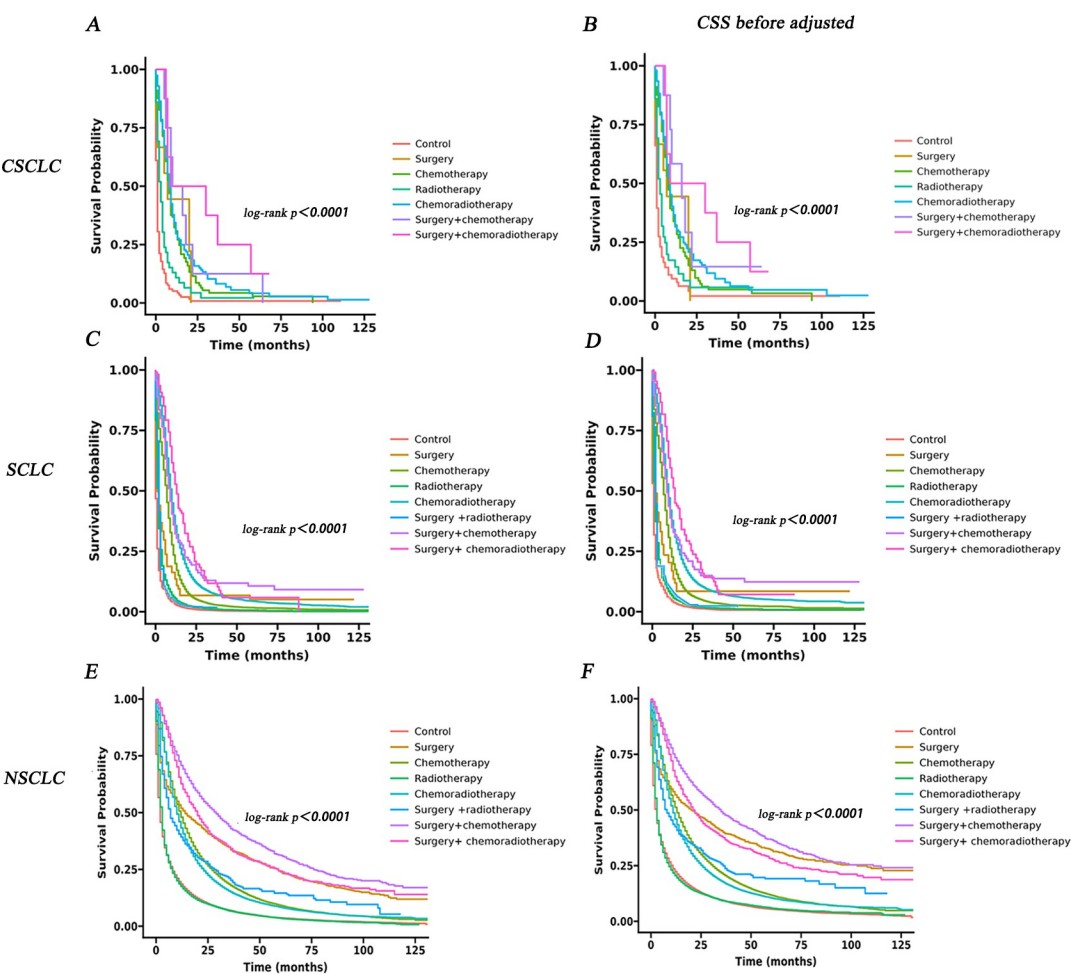

**Fig 11. KM curves for IV CSCLC, IV SCLC and IV NSCLC in different treatment modalities.** A, B KM curves of OS and CSS for IV CSCLC. C, D: KM curves of OS and CSS for IV SCLC. E, F: KM curves of OS and CSS for IV NSCLC.

**Table 6. The median survival (95% CI) of different treatment modalities in IV CSCLC.**

| therapy methods | CSCLC | | | |
|---|---|---|---|---|
| | mOS (95%CI) | P value* | mCSS (95%CI) | P Value* |
| Control | 1.0 (1.0,1.0) | — | 1.0 (1.0, 1.0) | — |
| Surgery | 7.0 (1.0, 9.0) | 0.004 | 7.0 (1.0, 9.0) | 0.026 |
| Chemotherapy | 8.0(7.0, 9.0) | <0.001 | 8.0 (7.0, 9.0) | <0.001 |
| Radiotherapy | 3.0 (2.0, 4.0) | <0.001 | 3.0 (2.0, 4.0) | <0.001 |
| Chemoradiotherapy | 8.0(7.0, 10.0) | <0.001 | 9.0 (8.0, 10.0) | <0.001 |
| Surgery+ chemotherapy | 13.0 (9.0, 15.0) | <0.001 | 16.0 (9.0,18.0) | <0.001 |
| Surgery + radiotherapy | — | — | — | — |
| Surgery+ chemoradiotherapy | 19.5(7.0, 25.0) | <0.001 | 19.5 (7.0, 25.0) | <0.001 |

*compared with control group, Log-rank

**Table 7. The median survival (95% CI) of different treatment modalities in IV SCLC.**

| therapy methods | SCLC | | | |
|---|---|---|---|---|
| | mOS (95%CI) | P value* | mCSS (95%CI) | P Value* |
| Control | 1.0 (1.0, 1.0) | — | 1.0 (1.0, 1.0) | — |
| Surgery | 2.0 (1.0, 3.0) | <0.001 | 2.0 (1.0, 4.0) | <0.001 |
| Chemotherapy | 7.0 (7.0, 7.0) | <0.001 | 7.0 (7.0, 7.0) | <0.001 |
| Radiotherapy | 2.0 (1.0, 2.0) | <0.001 | 2.0 (2.0, 2.0) | <0.001 |
| Chemoradiotherapy | 10.0 (9.0, 10.0) | <0.001 | 10.0 (10.0, 10.0) | <0.001 |
| Surgery+ chemotherapy | 9.0 (7.0, 11.0) | <0.001 | 9.0 (8.0, 11.0) | <0.001 |
| Surgery + radiotherapy | 2.0 (2.0, 3.0) | 0.008 | 2.0 (2.0, 5.0) | 0.006 |
| Surgery+ chemoradiotherapy | 13.0 (11.0, 17.0) | <0.001 | 14.0 (12.0, 17.0) | <0.001 |

*compared with control group, Log-rank

**Table 8. The median survival (95% CI) of different treatment modalities in IV NSCLC.**

| therapy methods | NSCLC | | | |
|---|---|---|---|---|
| | mOS (95%CI) | P value* | mCSS (95%CI) | P Value* |
| Control | 2.0 (2.0, 2.0) | — | 3.0 (2.0, 3.0) | — |
| Surgery | 14.0 (12.0, 17.0) | <0.001 | 20.0 (17.0, 24.0) | <0.001 |
| Chemotherapy | 12.0 (12.0, 12.0) | <0.001 | 13.0 (13.0, 13.0) | <0.001 |
| Radiotherapy | 3.0 (3.0, 3.0) | <0.001 | 3.0 (3.0, 3.0) | <0.001 |
| Chemoradiotherapy | 10.0 (10.0, 10.0) | <0.001 | 11.0 (11.0, 11.0) | <0.001 |
| Surgery+ chemotherapy | 29.0 (26.0, 32.0) | <0.001 | 35.0 (31.0, 40.0) | <0.001 |
| Surgery + radiotherapy | 7.0 (6.0, 9.0) | <0.001 | 8.0 (6.0, 11.0) | <0.001 |
| Surgery+ chemoradiotherapy | 20.0 (18.0, 22.0) | <0.001 | 21.0 (19.0, 25.0) | <0.001 |

*compared with control group, Log-rank

among the three groups. Thirdly, due to the limitations of diagnostic techniques and the low incidence and high mortality of CSCLC, we are unable to provide information on IV CSCLC patients from our own database. Future prospective studies are needed to further explore the relationship between CSCLC and NSCLC, SCLC.

## Conclusion

In summary, the clinical features of IV CSCLC differed from those of IV SCLC and IV NSCLC, the prognosis for patients with IV CSCLC was notably inferior to that of patients with IV NSCLC, but was similar to IV SCLC. Surgery, chemotherapy and radiotherapy all can improve survival time of IV CSCLC. Surgery combined with either chemoradiotherapy or chemotherapy emerged as the preferred treatment modalities for patients diagnosed with IV CSCLC, chemoradiotherapy was also a good choice for some IV CSCLC patients who have already lost the opportunity for surgery at the time of first diagnosis and chemotherapy regimen can referred to IV SCLC, which provide fresh insights and treatment strategies for the treatment of IV CSCLC patients.

## Supporting information

**S1 Fig. Standardized mean differences before and after PSM.** A: 1:4 match of surgery and non-surgical treatment groups; B: 1:1 match of radiotherapy and non-radiotherapy treatment

groups; C: 1:1 match of chemotherapy and non-chemotherapy treatment groups.
(TIF)

**S2 Fig. KM curves of IV CSCLC patients with each treatment modality before PSM.** A, B: KM curve of OS and CSS in surgery and non-surgical treatment groups. C, D: KM curve of OS and CSS in radiotherapy and non-radiotherapy treatment groups. E, F: KM curve of OS and CSS in chemotherapy and non-chemotherapy treatment groups.
(TIF)

**S3 Fig. KM curves for IV CSCLC, IV SCLC and IV NSCLC in different treatment modalities after adjusted.** A, B KM curves of OS and CSS for IV CSCLC. C, D: KM curves of OS and CSS for IV SCLC. E, F: KM curves of OS and CSS for IV NSCLC.
(TIF)

**S1 Table. Univariable Cox analysis of OS and CSS in IV CSCLC, SCLC and NSCLC before PSM.**
(DOCX)

**S2 Table. Univariable Cox analysis of OS and CSS in IV CSCLC, SCLC and NSCLC after 1:4 PSM.**
(DOCX)

**S3 Table. The baseline data of patients undergoing surgical (No/Yes) intervention for IV CSCLC before and after PSM.**
(DOCX)

**S4 Table. The baseline data of patients undergoing radiotherapy (No/Yes) intervention for IV CSCLC before and after PSM.**
(DOCX)

**S5 Table. The baseline data of patients undergoing chemotherapy(No/Yes) intervention for IV CSCLC before and after PSM.**
(DOCX)

**S6 Table. The baseline characteristics of different treatment modalities in IV CSCLC.**
(DOCX)

**S7 Table. The baseline characteristics of different treatment modalities in IV SCLC.**
(DOCX)

**S8 Table. The baseline characteristics of different treatment modalities in IV NSCLC.**
(DOCX)

**S9 Table. Cox regression for efficacy analysis of IV CSCLC.** * OS adjusted for Age(years), Race, N stage, Bone Metastasis, and Liver Metastasis, *CSS adjusted for Age(years), Race, T stage, N stage, Bone Metastasis, and Liver Metastasis.
(DOCX)

**S10 Table. Cox regression for efficacy analysis of IV CSCLC.** * OS and CSS adjusted for for Age(years), Gender, Race, Married status, Primary Site, T stage, N stage, Bone Metastasis, Brain Metastasis, Liver Metastasis, and Lung Metastasis.
(DOCX)

**S11 Table. Cox regression for efficacy analysis of IV CSCLC.** * OS and CSS adjusted for Age years, Gender, Race, Married status, Primary Site, T stage, N stage, Bone Metastasis, Brain

Metastasis, Liver Metastasis, and Lung Metastasis.
(DOCX)

## Author Contributions

**Data curation:** Hongdan Luo.

**Funding acquisition:** Xiaoqun Ye.

**Project administration:** Xiaotian Huang.

**Resources:** Weichang Yang, Zhouhua Li.

**Software:** Jinbo Li, Xiaoqun Ye.

**Supervision:** Xiaoqun Ye.

**Validation:** Xiaoqun Ye.

**Writing – review & editing:** Shanshan Cai.

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
