## [Decision Letter · Decision Letter 0]

22 Aug 2024

PONE-D-24-30493

Clinical features and prognostic factors of IV combined small cell lung cancer: A Propensity Score Matching Analysis

PLOS ONE

Dear Dr. Ye,

Thank you for submitting your manuscript to PLOS ONE. After careful consideration, we feel that it has merit but does not fully meet PLOS ONE’s publication criteria as it currently stands. Therefore, we invite you to submit a revised version of the manuscript that addresses the points raised during the review process.

We look forward to receiving your revised manuscript.

Kind regards,

Luca Bertolaccini, M.D., Ph.D.

Academic Editor

PLOS ONE

Journal Requirements:

6. Please amend the manuscript submission data (via Edit Submission) to include author Zhouhua Li , Xiaotian Huang and Jinbo Li.

7. We note you have included a table to which you do not refer in the text of your manuscript. Please ensure that you refer to Table 5 in your text; if accepted, production will need this reference to link the reader to the Table.

Additional Editor Comments:

The reviewers have emphasised issues that require a careful and thorough manuscript revision.

No commitment to publication can be made at this point.

Reviewers' comments:

Reviewer's Responses to Questions

**Comments to the Author**

1. Is the manuscript technically sound, and do the data support the conclusions?

Reviewer #1: No

Reviewer #2: Partly

2. Has the statistical analysis been performed appropriately and rigorously? 

Reviewer #1: No

Reviewer #2: I Don't Know

3. Have the authors made all data underlying the findings in their manuscript fully available?

Reviewer #1: Yes

Reviewer #2: No

4. Is the manuscript presented in an intelligible fashion and written in standard English?

Reviewer #1: Yes

Reviewer #2: No

5. Review Comments to the Author

Reviewer #1: 1. Instead of using performance status or comorbidities, the authors used initial treatment and metastatic status as variables for PSM. These variables could be confounding, as the initial treatment may not be independent of metastatic status.

2. Regarding the Kaplan-Meier analysis, the results of "Before PSM" and "After PSM" are mixed within the chapter, making it difficult for readers to understand. Please separate the sections for "Before PSM" and "After PSM" in the chapter for both patient characteristics and KM analysis. Additionally, the authors are conducting two analyses: CSCLC vs. SCLC and CSCLC vs. NSCLC. Since the CSCLC populations in each analysis are different, the authors should not present the results of three cohorts in parallel within the same sections, as this could confuse readers and lead to misleading conclusions.

3. Patients who received radiotherapy as the initial treatment in stage IV were symptomatic and received palliative radiotherapy, typically associated with poor prognosis in clinical practice. In contrast, those who underwent surgery were likely patients with relatively localized metastatic disease or in good PS. This could introduce significant bias in comparisons, so PSM is usually employed for such comparisons. However, the main focused PSM of this study is on CSCLC vs. SCLC and CSCLC vs. NSCLC, and another PSM is required to demonstrate the authors’ hypothesis. The sections on "the prognosis of each treatment modality in IV CSCLC patients" and "the evaluation of different treatment modalities" would be aside from the main objectives in this study, and the statistical method is not appropriate, even though it is a subsequent analysis. The authors should reconsider the main objectives and results of this study in conjunction with the statistical methods used.

Reviewer #2: Cai et al. investigated Clinical features and prognostic factors of IV combined small cell lung cancer: A Propensity Score Matching Analysis, They indicated that the prognosis and clinical characteristics IV CSCLC, IV SCLC and IV NSCLC were significant difference. Surgery combined chemotherapy was the best treatment in patients with IV CSCLC and chemotherapy alone was a good choice for patients who have lost the indication of surgery.However following issued shoud be addressed:

1 In table 1 before PSM ,there are so many P-value <0.05 ,but in table 2 after PSM ,there are less sample size ,which will cause bias,please explain it.

2 all figs are low quality,please improve the absolution.

3 the conclusion should sumarize more concisely and logicaly. you indicate "we found that the prognosis of IV CSCLC patients was significantly worse

than that of IV NSCLC patients, but better than IV SCLC patients." please provide evidence.

4 the format of study should revise accorading scientific format.please check it.

5 Table 5 1: The mOS and mCSS of different therapy methods in IV CSCLC, SCLC and IV SCLC,there is no statistic analysis to show in table ,you should compare to analysize to achieve the conclusion.

6. PLOS authors have the option to publish the peer review history of their article (what does this mean?). If published, this will include your full peer review and any attached files.

Reviewer #1: No

Reviewer #2: No

---

## [Author Response · Author response to Decision Letter 0]

23 Sep 2024

Dear esteemed Editor and Reviewers,

I am writing to express my sincere gratitude for the invaluable time and effort you have dedicated to reviewing my manuscript “Clinical features and prognostic factors of IV combined small cell lung cancer: A Propensity Score Matching Analysis" (manuscript number: PONE-D-24-30493)”. Your insightful comments and suggestions have been instrumental in enhancing the quality and clarity of my work.

I have thoroughly revised the manuscript in accordance with your guidelines and recommendations. In addition to addressing each of your specific suggestions, I have also conducted a careful review for grammatical errors, further refining the text for improved readability. To facilitate your review of the changes, I have highlighted the corrections for grammatical errors in yellow and marked the revisions that directly respond to the reviewers' comments in red. Our responses are outlined below, and the corrections made to the manuscript are explained.

Editor Requirements:

Reply: Thanks to your valuable suggestion, I have reformatted the manuscript to conform precisely to PLOS ONE's specifications, including adjustments to font size, margins, line spacing, reference format, etc. The file naming convention has been strictly followed for both the manuscript and any accompanying materials.

Reply: Thank you for your advice. I have deleted the funding information part in the manuscript.

3. We note that the grant information you provided in the ‘Funding Information’and ‘Financial Disclosure’ sections do not match. When you resubmit, please ensure that you provide the correct grant numbers for the awards you received for your study in the ‘Funding Information’ section.

Reply: Thank you for your careful review. We apologize for our previous oversight. we modified Funding Information and Financial Disclosure.

Reply: Thanks, I have provided a complete Data Availability Statement in the submission form.

Reply: Thanks, I have included the ORCID ID of the corresponding author to enhance discoverability and credit attribution.

6. Please amend the manuscript submission data (via Edit Submission) to include author Zhouhua Li , Xiaotian Huang and Jinbo Li.

Reply: Thanks, I have reviewed and updated authors information to ensure its accuracy and completeness, adhering to PLOS ONE's requirements.

7. We note you have included a table to which you do not refer in the text of your manuscript. Please ensure that you refer to Table 5 in your text; if accepted, production will need this reference to link the reader to the Table.

Reply: Thank you for your reminder. After substantial revisions to the full text, I have carefully checked it to ensure that such an issue will not arise. All figures and tables have been properly cited.

Reviewer's comments:

Reviewer #1

1. Instead of using performance status or comorbidities, the authors used initial treatment and metastatic status as variables for PSM. These variables could be confounding, as the initial treatment may not be independent of metastatic status.

Reply: Thank you for your suggestions. In this study, I have incorporated the performance status (including age, gender, race, marital status, laterality, and primary site) and stage (T stage, N stage) as per your request. However, after meticulous contemplation and thorough literature review, I discovered that both the metastatic states (including brain, bone, liver, and lung metastasis) and the initial treatments (surgery, radiotherapy, chemotherapy), along with the aforementioned balancing variables[1-12] can significantly impact the survival outcomes of lung cancer patients. Our objective in PSM is to ensure the comparability of survival outcomes across the three patient groups. Given that these factors exert a pivotal influence on patients' survival, failing to include them in the analysis could potentially introduce substantial bias into subsequent statistical comparisons. Indeed, my subsequent univariate and multivariate Cox regression analyses on the primary survival outcomes further validated that the metastatic status and initial treatments are independent risk and protective factors, respectively, for the three groups. Furthermore, upon reviewing the literature, I found that previous studies in this domain consistently incorporated the metastatic status and initial treatment as influential variables within their PSM analyses[13-17]. Consequently, based on these considerations，I have decided to include these two types of variables in my own analysis as well. All the references mentioned above are attached at the end of this paper for further verification and reference.

2. Regarding the Kaplan-Meier analysis, the results of "Before PSM" and "After PSM" are mixed within the chapter, making it difficult for readers to understand. Please separate the sections for "Before PSM" and "After PSM" in the chapter for both patient characteristics and KM analysis. 

Reply: Thank you sincerely for your insightful suggestions. I have segregated the results into distinct sections, clearly differentiating between those obtained before and after PSM. Furthermore, I have meticulously redrawn the survival curves and segregated them into two distinct figures (Figure 4 and Figure 5) to clearly illustrate the before PSM and after PSM survival outcomes among three groups. To facilitate easy identification, the revised content has been conspicuously highlighted in red font throughout the text.

3. Additionally, the authors are conducting two analyses: CSCLC vs. SCLC and CSCLC vs. NSCLC. Since the CSCLC populations in each analysis are different, the authors should not present the results of three cohorts in parallel within the same sections, as this could confuse readers and lead to misleading conclusions.

Reply: Thank you for your reminder. To mitigate bias and ensure the rigorousness of our research, I have revised the study design to adopt a 1:4 PSM, significantly expanding the sample sizes for both IV NSCLC and IV SCLC（Tables 2,3 and Fig 2）. This adjustment included all CSCLC samples (n=493) and the problem of different CSCLC populations in each analysis was solved. 

During the analysis of risk factors, because the population of CSCLC is the same every time, isolating CSCLC against SCLC and NSCLC individually would result in redundant occurrences of the same CSCLC outcomes, unnecessarily complicating the process. Consequently, so the three groups are presented within a single table for clarity (Table 5). Nevertheless, in the subsequent result interpretation, apart from some influencing factors shared by the three groups, we also conduct distinct analyses comparing CSCLC versus SCLC and CSCLC versus NSCLC, respectively, the revised content has been conspicuously highlighted in red font throughout the text and table.

4. Patients who received radiotherapy as the initial treatment in stage IV were symptomatic and received palliative radiotherapy, typically associated with poor prognosis in clinical practice. In contrast, those who underwent surgery were likely patients with relatively localized metastatic disease or in good PS. This could introduce significant bias in comparisons, so PSM is usually employed for such comparisons. However, the main focused PSM of this study is on CSCLC vs. SCLC and CSCLC vs. NSCLC, and another PSM is required to demonstrate the authors’ hypothesis. The sections on " the prognosis of each treatment modality in IV CSCLC patients" and "the evaluation of different treatment modalities" would be aside from the main objectives in this study, and the statistical method is not appropriate, even though it is a subsequent analysis. The authors should reconsider the main objectives and results of this study in conjunction with the statistical methods used.

Reply: Thank you sincerely for your insightful suggestions. I have adjusted statistical method for the two sections, the corresponding statistical methods, results and discussion sections have been revised too. The revised content has been conspicuously highlighted in red font throughout the text and table.

For the section on “the prognosis of each treatment modality in IV CSCLC patients”, PSM was applied to each treatment modality in IV CSCLC patients in this study per your request. For surgery vs. no surgery groups, a 1:4 matching ratio with a 0.02 caliper value was adopted using the “nearest” method (S1A Fig, S3 Table). Similarly, 1:1 ratio and 0.02 caliper values were applied to radiotherapy vs. no radiotherapy groups (S1B Fig, S4 Table) and chemotherapy vs. no chemotherapy groups (S1C Fig, S5 Table), also utilizing the "nearest" method. All covariates were subsequently well balanced (p≥0.05 and SMD＜0.1). Matching variables encompassed age, gender, race, marital status, T stage, N stage, laterality, primary site, brain metastasis, bone metastasis, liver metastasis, lung metastasis, along with other therapies received (surgery /radiotherapy/ chemotherapy). In addition, we also added the survival curve after PSM (Fig 10)

For the section on “Evaluation of different treatment modalities of IV CSCLC, IV SCLC and IV NSCLC”, For comparison among baseline characteristics of different treatment modalities, we used the Fisher exact test for expected frequencies of <5, otherwise, we used the Chi-squared test. Kaplan-Meier method and log-rank test were used to compare the prognosis of different treatment modalities, we revised the table about mOS and mCSS of different therapy methods in three groups, the median survival (95% CI) and log-rank P value compared with control group were included in the tables 6-8. In addition, Adjusted analyses for the primary survival outcome of different treatment modalities of IV CSCLC, IV SCLC and IV NSCLC were performed using Cox regression models to estimated HRs with corresponding two sides 95CIs, considering potential unbalanced confounders that may have influenced the outcomes (S9-S11 Tables). The survival curve after adjusted was shown in S3 Fig. The covariates included in the adjustment were confounding factors that were found by cox regression analysis to be likely to influence prognosis. The adjusted covariates in each group were described in each table legend. 

Reviewer #2

1. In table 1 before PSM, there are so many P value <0.05, but in table 2 after PSM, there are less sample size, which will cause bias, please explain it.

Reply: Thank you sincerely for your insightful suggestions. To mitigate potential biases and ensure the rigorousness of our research. I have revised the study design to change 1:1 PSM to 1:4 PSM, significantly expanding the sample sizes for both IV NSCLC and IV SCLC. All covariates were subsequently well balanced both in the 1:4 matched cohort of IV CSCLC (n=493) vs IV SCLC (n=1954), and the 1:4 matched cohort of IV CSCLC (n=493) vs IV NSCLC (n=1968). (p≥0.05 and SMD＜0.1, Table2,3, Fig 2). We also modified the subsequent research results accordingly; the revised content has been conspicuously highlighted in red font throughout the text and table.

2. all figs are low quality, please improve the absolution.

Reply: Thank you for your suggestion. I have redrawn all the figures and ensured that they meet the journal requirements of PLOS ONE by maintaining a resolution of above 300 ppi. All figures have been uploaded in TIF format.

3. the conclusion should summarize more concisely and logically. you indicate "we found that the prognosis of IV CSCLC patients was significantly worse than that of IV NSCLC patients, but better than IV SCLC patients." please provide evidence.

Reply: Thanks, I have revised the section of conclusion and corrected the imprecise statement as your request with highlighted in red. Importantly, I have revised “we found that the prognosis of IV CSCLC patients was significantly worse than that of IV NSCLC patients, but better than IV SCLC patients” to “the prognosis for patients with IV CSCLC was notably inferior to that of patients with IV NSCLC, but was similar to IV SCLC”. This conclusion is elaborated upon in detail in section “KM analysis for IV CSCLC, IV SCLC and IV NSCLC before PSM” and section “KM analysis for IV CSCLC, IV SCLC and IV NSCLC after PSM”, with modifications highlighted in red.

4. the format of study should revise according scientific format. Please check it.

Reply: Thanks to your valuable suggestion, I have reformatted the manuscript to conform precisely to PLOS ONE's specifications, including adjustments to font size, margins, line spacing, reference format, etc. Tables and figures were also redrawn as required. In addition, I have also conducted a careful review for grammatical errors, which highlighted in yellow.

5. Table 5: The mOS and mCSS of different therapy methods in IV CSCLC, SCLC and IV SCLC, there is no statistic analysis to show in table, you should compare to analysize to achieve the conclusion.

Reply: Thanks to your valuable suggestion. In our study, Kaplan-Meier method and log-rank test were used to compare the prognosis of different treatment modalities, we revised the table about mOS and mCSS of different therapy methods in three groups per your request, the median survival (95% CI) and log-rank P value compared with control group were included in the tables 6-8. In addition, adjusted analyses for the primary survival outcome of different treatment modalities of IV CSCLC, IV SCLC and IV NSCLC were performed using Cox regression models to estimated HRs with corresponding two sides 95CIs, considering potential unbalanced confounders that may have influenced the outcomes (S9-S11 Tables). The revised content has been conspicuously highlighted in red font throughout the text and table.

Thanks again to the Editor and Reviewers for their valuable comments. We now resubmit the revised manuscript and hope that all corrections are satisfactory. We look forward to hearing from you regarding our submission. We would be glad to respond to any further questions and comments that you may have.

Best wishes!

Yours Sincerely

Xiaoqun Ye

Reference

1. Hemminki K, Riihimäki M, Sundquist K, Hemminki A. Site-specific survival rates for cancer of unknown primary according to location of metastases. International journal of cancer. 2013;133(1):182-9. Epub 2012/12/13. doi: 10.1002/ijc.27988. PubMed PMID: 23233409.

2. Jakobsen E, Rasmussen TR, Green A. Mortality and survival of lung cancer in Denmark: Results from the Danish Lung Cancer Group 2000-2012. Acta oncologica (Stockholm, Sweden). 2016;55 Suppl 2:2-9. Epub 2016/04/09. doi: 10.3109/0284186x.2016.1150608. PubMed PMID: 27056247.

3. Kehl KL, Lathan CS, Johnson BE, Schrag D. Race, Poverty, and Initial Implementation of Precision Medicine for Lung Cancer. Journal of the National Cancer Institute. 2019;111(4):431-4. Epub 2018/12/24. doi: 10.1093/jnci/djy202. PubMed PMID: 30576459; PubMed Central PMCID: PMCPMC6449167.

4. Meza R, Meernik C, Jeon J, Cote ML. Lung cancer incidence trends by gender, race and histology in the United States, 1973

---

## [Decision Letter · Decision Letter 1]

22 Oct 2024

Clinical features and prognostic factors of IV combined small cell lung cancer: A Propensity Score Matching Analysis.

PONE-D-24-30493R1

Dear Dr. Ye,

We’re pleased to inform you that your manuscript has been judged scientifically suitable for publication and will be formally accepted for publication once it meets all outstanding technical requirements.

Kind regards,

Luca Bertolaccini, M.D., Ph.D.

Academic Editor

PLOS ONE

Additional Editor Comments (optional):

Reviewers' comments:

Reviewer's Responses to Questions

**Comments to the Author**

1. If the authors have adequately addressed your comments raised in a previous round of review and you feel that this manuscript is now acceptable for publication, you may indicate that here to bypass the “Comments to the Author” section, enter your conflict of interest statement in the “Confidential to Editor” section, and submit your "Accept" recommendation.

Reviewer #2: All comments have been addressed

Reviewer #3: All comments have been addressed

2. Is the manuscript technically sound, and do the data support the conclusions?

Reviewer #2: (No Response)

Reviewer #3: Yes

3. Has the statistical analysis been performed appropriately and rigorously? 

Reviewer #2: Yes

Reviewer #3: Yes

4. Have the authors made all data underlying the findings in their manuscript fully available?

Reviewer #2: Yes

Reviewer #3: Yes

5. Is the manuscript presented in an intelligible fashion and written in standard English?

Reviewer #2: Yes

Reviewer #3: Yes

6. Review Comments to the Author

Reviewer #2: (No Response)

Reviewer #3: Clinical features and prognostic factors of IV combined small cell lung cancer: A Propensity Score Matching Analysis

Thank you for asking me to review this manuscript.

The authors of the article deal with a very interesting topic: CSCLC Combined small-cell lung cancer, a rare tissue type of lung cancer.

This disease combines the two best known histological subtypes within the same neoplasia. It is an uncommon condition so we still have a lot to learn about this particular form of neoplasia.

The authors’ significant retrospective analysis was based on the data of the SEER database and their research strategy involved the Propensity score matching (PSM) that was chosen to balance the bias of the variables between patients.

The article is well structured, the data are clearly presented and the results are well explained.

From the conclusions we gather that this rare form of pulmonary carcinoma has a prognosis that we can consider in between the two kinds that are best known to the experts of this field. To be more precise, the research shows us that it is much closer to the prognosis of the SCLC: this suggests that the presence of the kind of cells that cause the SCLC variant of the tumor has a negative impact on the prognosis of the combined variant.

This interesting research shows also that a multimodal treatment that includes surgery is the best therapeutic approach in these cases and that chemoradiotherapy is a valuable and effective option when surgery is no longer a possibility for the patient.

The authors themselves point out that their research has some weaknesses such as the fact that the SEE database could be improved and the fact that molecular analysis were not included in their study.

I believe this research is an important starting point that could eventually be developed into a prospective study in order to better investigate this rare isotope of pulmonary cancer and learn how to treat it best.

7. PLOS authors have the option to publish the peer review history of their article (what does this mean?). If published, this will include your full peer review and any attached files.

Reviewer #2: No

Reviewer #3: No

---

## [Editor Report · Acceptance letter]

29 Oct 2024

PONE-D-24-30493R1 

PLOS ONE

Dear Dr. Ye, 

I'm pleased to inform you that your manuscript has been deemed suitable for publication in PLOS ONE. Congratulations! Your manuscript is now being handed over to our production team.

Kind regards, 

on behalf of

Dr. Luca Bertolaccini 

Academic Editor

PLOS ONE